# Success Conditioning as Policy Improvement:
# The Optimization Problem Solved by Imitating Success

**Daniel J. Russo** [1]

## Abstract

A widely used technique for improving policies is success conditioning, in which one collects trajectories, identifies those that achieve a desired outcome, and updates the policy to imitate the actions taken along successful trajectories. This principle appears under many names—rejection sampling with SFT, goal-conditioned RL, Decision Transformers—yet what optimization problem it solves, if any, has remained unclear. We prove that success conditioning exactly solves a trust-region optimization problem, maximizing policy improvement subject to a $\chi^2$ divergence constraint whose radius is determined automatically by the data. This yields an identity: relative policy improvement, the magnitude of policy change, and a quantity we call action-influence—measuring how random variation in action choices affects success rates—are exactly equal at every state. Success conditioning thus emerges as a conservative improvement operator. Exact success conditioning cannot degrade performance or induce dangerous distribution shift, but when it fails, it does so observably, by hardly changing the policy at all. We apply our theory to the common practice of return thresholding, showing this can amplify improvement, but at the cost of potential misalignment with the true objective.

## 1. Introduction

We study success conditioning: a heuristic for improving policies in which one collects trajectories, identifies those that achieve a desired outcome, and updates the policy to imitate the actions taken along successful trajectories. This principle aims to improve performance while sidestepping explicit policy optimization, and appears across domains.

[1]Columbia Business School, New York, NY, USA. Correspondence to: Daniel Russo <djr2174@columbia.edu>.

*Proceedings of the 43rd International Conference on Machine Learning*, Seoul, South Korea. PMLR 306, 2026. Copyright 2026 by the author(s).

In language model alignment, rejection sampling followed by supervised fine-tuning on high-reward completions is now a core component of post-training pipelines (Touvron et al., 2023; Grattafiori et al., 2024). In reasoning, methods like STaR fine-tune on self-generated rationales that yield correct answers (Zelikman et al., 2022), while large-scale systems such as DeepSeek-R1 apply the same strategy to stabilize reinforcement learning (Guo et al., 2025). In tool use, models are trained on calls that improve downstream predictions (Schick et al., 2023), and in embodied learning, success detectors filter autonomously collected trajectories (Ghasemipour et al., 2025). In a distinct literature within reinforcement learning, the same idea appears under a different guise: return-conditioned models such as Decision Transformers generate actions conditional on achieving high returns (Schmidhuber, 2019; Chen et al., 2021).

Despite differing implementations, these methods share a common structure: *learn to act as you would have, given that you eventually succeed.* Intuitively, this might improve upon the behavior policy by retaining what worked and discarding what did not. Yet it is unclear how to understand success conditioning in a principled manner, as it does not estimate value functions, compute policy gradients, or otherwise perform explicit policy optimization. The abstract of the original Decision Transformer paper (Chen et al., 2021) emphasizes this contrast, stating informally: "Unlike prior approaches to RL that fit value functions or compute policy gradients, Decision Transformer simply outputs the optimal actions by leveraging a causally masked Transformer."

Subsequent work examined this interpretation and showed that, when viewed as a route to optimal policy recovery, success conditioning has fundamental limitations, particularly in stochastic environments (Brandfonbrener et al., 2022; Paster et al., 2022; Yang et al., 2023). Example 3.1 presents a simple two-armed bandit in which a single step of success conditioning yields negligible improvement.

At the same time, success conditioning continues to see widespread use across reinforcement learning and post-training pipelines. This raises a natural question: *if success conditioning does not recover optimal policies, what is it optimizing for?*

## 1.1. Our Contribution

We show that success conditioning can be precisely interpreted as a *conservative policy improvement operator*: one that finds the largest gain achievable without venturing too far outside observed behavior. The most notable failures of success conditioning have a distinctive character that aligns exactly with this interpretation. Proper success conditioning (defined by $\pi_+$ in Section 2) does not fail unpredictably at deployment time by moving to under-performing actions or inducing harmful distribution shift. Instead, when it fails, it does so by barely changing the policy at all—a form of over-conservatism. The boxes below summarize our primary novel contributions:

---

### The Trust-Region Equivalence

Success conditioning is the exact solution to a trust-region policy optimization problem with unique geometry—$\chi^2$ divergence rather than KL—and a radius determined automatically by a quantity we call action-influence, which measures the variability in success rates induced by the behavior policy's action selection.

---

### The Action-Influence Identity

The following are equal at every state:

1. Relative improvement from success conditioning, as measured by expected relative advantage

2. Policy change from the behavior policy, as measured by $\chi^2$ divergence

3. Action-influence under the behavior policy

These quantities are not merely related but exactly equal, implying that the extent of policy change—and equivalently, the variability induced by action selection under the behavior policy—provides a direct diagnostic of the magnitude of improvement.

---

These results place success conditioning in the same family as TRPO and PPO (Schulman et al., 2015; 2017)—trust-region methods that maximize a local approximation to performance while remaining close to the current policy. However, success conditioning differs from these methods in three fundamental ways.

First, success conditioning is typically not formulated or implemented as a trust-region optimization procedure: it is defined via Bayesian conditioning and implemented by training sequence models on filtered trajectories. Second, the implicit trust region induced by success conditioning has a radius that is determined automatically by variation in the

offline data, rather than by a user-chosen hyperparameter. Third, the geometry of the trust region is substantially more conservative. The $\chi^2$ divergence severely restricts exploration outside the support of the behavior policy, in contrast to the KL-based trust regions used in TRPO and PPO (see Section 4.5).

**Outline of the paper.** Section 2 formalizes success conditioning in the setting of episodic Markov decision processes and defines the success-conditioned policy $\pi_+$. Section 4 contains our main theory, showing that success conditioning exactly solves a trust-region policy optimization problem and establishing the action-influence identity. Section 6 discusses the implications of this perspective for learning $\pi_+$ from data and for the reliability of supervised fine-tuning on successful trajectories. Section 7 discusses success conditioning in problems where rewards are dense. It interprets the common practice of return conditioning as optimizing a proxy reward that may amplify action-influence but risks misalignment with the true objective. Section 8 situates our results within the broader literature.

## 2. Markov Decision Process Formulation

We formulate the problem using the language of Markov Decision Processes. Later, we will see how examples like language model generation can fit this model.

A trajectory in an episodic Markov decision process is a sequence $\tau = (S_1, A_1, \ldots, S_{T-1}, A_{T-1}, S_T)$ consisting of discrete states $S_t \in \mathcal{S}$ and actions $A_t \in \mathcal{A}$. The episode ends at the first time $T$ at which $S_t \in \mathcal{T}$, where $\mathcal{T} \subset \mathcal{S}$ is a set of terminal states. Conditioning on success is inherently conditioning on a binary event. As such, we formalize success events by partitioning $\mathcal{T}$ into successful and failed terminal states, and define a binary reward $R(\tau) \in \{0, 1\}$ indicating whether the trajectory ended in success. See Section 7 for a precise reduction from general bounded rewards and a discussion of proxy success criteria, including return thresholding for continuous rewards.

Observed trajectories are generated by a Markovian **behavior policy** $\pi_0$. We have that $S_1$ is drawn from a fixed initial distribution $\mu$ and, at any time $t$, the next action is drawn conditioned on the current state as $A_t \sim \pi_0(\cdot \mid S_t)$ and transitions follow a fixed kernel $S_{t+1} \sim P(\cdot \mid S_t, A_t)$.

**Success conditioning.** From observed trajectories and rewards, we define the success-conditioned policy

$$\pi_+(a \mid s) = P_{\pi_0}\left(A_t = a \mid S_t = s, R(\tau) = 1\right).$$

This policy mimics the distribution of actions taken along trajectories that ended in success. To avoid repeated qualifiers regarding division by zero, we assume that success

is possible from any non-terminal state under $\pi_0$, though it may occur with arbitrarily small probability.

Note that the success-conditioned policy can be viewed as the minimizer of the next-action-prediction loss on successful trajectories,

$$\pi_+ \in \arg\min_{\hat{\pi}} \left\{ \ell(\hat{\pi}) := \mathbb{E}\left[ -\sum_{t=1}^{T-1} \log \hat{\pi}(A_t|S_t) \right], \right\}$$
$$\text{where } (S_1, A_1, \ldots, S_T) \sim P_{\pi_0}(\tau \mid R(\tau) = 1)$$

and approximated by training a sequence model on a dataset of successful trajectories. We mainly focus on exact success conditioning, touching on approximation errors in Prop. 6.1.

**Example 2.1** (Language Model Fine-Tuning)**.** In language generation, a state $S_t$ is the sequence of tokens generated so far, an action $A_t \in \mathcal{V}$ is the next token selected from vocabulary $\mathcal{V}$ and the transition is deterministic: tokens are appended as $S_{t+1} = (S_t, A_t)$. The behavior policy $\pi_0$ is the base model's next-token distribution. Success $R(\tau) = 1$ indicates the completed sequence met some criterion—for instance, a correct answer or a positive human rating. Supervised fine-tuning on successful completions trains a model to approximate $\pi_+$: it imitates the token-level choices made along trajectories that ended in success.

*Remark* 2.1. The state $S_t$ may encode the full history of observations up to time $t$. This is necessary when the transition kernel depends on history, but also when $\pi_0$ is not itself a single Markov policy—for instance, if trajectories are collected from a mixture of policies. Architectures like transformers make it feasible to operate with history as state.

### 2.1. Some Notation

**We follow the convention throughout that $s$ denotes a non-terminal state. Hence, the event $S_t = s$ already implies that $T > t$ and the episode has not yet terminated.**

We define a few key terms:

- Success probability: $\rho(\pi) := P_\pi(R(\tau) = 1)$

- Value function: $V_\pi(s) = P_\pi(R(\tau) = 1 \mid S_t = s)$

- Q-value: $Q_\pi(s, a) = P_\pi(R(\tau) = 1 \mid S_t = s, A_t = a)$

- Advantage function: $A_\pi(s, a) = Q_\pi(s, a) - V_\pi(s)$

- **Special notation:** $A_\pi(s, \pi') := \mathbb{E}_{a \sim \pi'(s, \cdot)}[A_\pi(s, a)]$.

- Occupancy measure: $d_\pi(s) = \mathbb{E}_\pi\left[ \sum_{t=1}^{T-1} \mathbb{1}(S_t = s) \right]$

- Success conditioned occupancy $d_\pi^+(s) = \mathbb{E}_\pi\left[ \sum_{t=1}^{T-1} \mathbb{1}(S_t = s) \mid R(\tau) = 1 \right]$.

Among this notation, the most atypical is the success-conditioned occupancy $d_\pi^+$. A subtle but important point—and a source of potential errors—is conflating this with $d_{\pi^+}$, the occupancy measure under the success-conditioned policy. The former conditions on success, selecting both for the actions taken and for "lucky draws" from the environment along the way. The latter imitates those actions, but cannot re-create the luck. These two measures coincide in deterministic environments, such as the language generation setting of Example 2.1.

## 3. Action-Influence and the Failure Mode of Success Conditioning

### 3.1. The Failure Mode of Success Conditioning

**Example 3.1.** Consider a two-armed bandit. There is no state and the process terminates after a single period. Arm 1 succeeds with probability 0.495 and arm 2 with probability 0.505. The behavior policy pulls each arm with equal probability, achieving an overall success rate of 0.5.

The success-conditioned policy is easy to compute. By Bayes' rule, it assigns probability $\pi_+(a) = P(R = 1 \wedge A_1 = a)/P(R = 1)$ to each arm: 0.495 to arm 1 and 0.505 to arm 2. This improves the success rate from 0.5 to 0.50005—a gain of 0.00005.

From sufficient data, it would be easy to learn the optimal policy, which pulls arm 2 always and has success rate 0.505. Success conditioning closes only 1% of the gap to optimal.

What went wrong? Success conditioning asks: conditioned on success, which arm was pulled? Since both arms succeed at nearly the same rate, the posterior barely differs from the prior. **Due to low variation in success rates across actions, the success-conditioned policy remains nearly identical to the behavior policy, offering minimal improvement**. Unlike RL methods that fail due to updating the policy dangerously—resulting in uncontrolled distribution shift—success conditioning seems to fail observably by hardly updating the policy at all.

### 3.2. Influence of the Action-Draw

The example above hints that the extent of improvement offered by success conditioning is governed by some notion of variability in action-success. The right measure of this turns out to be the measure below.

**Definition 3.1.** The influence of the action-draw at a state is defined by the square of the *coefficient of variation* of the $Q_{\pi_0}(s, \cdot)$ over the action draw:

$$\mathcal{I}_{\pi_0}(s) := \left( \frac{\text{Stdev}_{a \sim \pi_0(\cdot|s)}[Q_{\pi_0}(s, a)]}{\mathbb{E}_{a \sim \pi_0(\cdot|s)}[Q_{\pi_0}(s, a)]} \right)^2$$

We abbreviate it as the action-influence at state $s$. It measures statistical variation in success probabilities induced by the stochastic action draw under the behavior policy.

*Remark* 3.2. It is also possible to show the alternate form,

$$\mathcal{I}_{\pi_0}(s) = \mathbb{E}_{a \sim \pi_0(\cdot|s)} \left[ \left( \frac{A_{\pi_0}(s,a)}{V_{\pi_0}(s)} \right)^2 \right].$$

Notice that action-influence is zero if the behavior policy is deterministic or if no two actions result in substantially different success probabilities. It is large only when the stochasticity in the behavior policy meaningfully affects final outcomes. In Example 3.1, $\mathcal{I}_{\pi_0} = 0.5 \times \left( \frac{-0.005}{0.5} \right)^2 + 0.5 \times \left( \frac{0.005}{0.5} \right)^2 = 0.0001$, precisely capturing the limited signal provided by a success indicator.

# 4. Success Conditioning as Trust Region Optimization

## 4.1. Background on Policy Improvement Theory

Given a behavior policy $\pi_0$ how do we find a new policy $\pi$ with improved success probability, i.e. $\rho(\pi) > \rho(\pi_0)$. Classical dynamic programming theory suggests to find a *single period deviation* from $\pi_0$, by seeking a positive one-step advantage ($A_{\pi_0}(s, \pi) > 0$) at all states. A policy gradient perspective suggests instead to improve $\pi_0$ through *small, but persistent policy deviations*.

These two views turn out to be intimately linked. The performance difference lemma (Kakade & Langford, 2002) enables us to expand $\rho(\pi)$ around the behavior policy $\pi_0$ as:

$$\rho(\pi) = \rho(\pi_0) + L_{\pi_0}(\pi) + \text{Rem}_{\pi_0}(\pi)$$

where the first-order term

$$L_{\pi_0}(\pi) := \sum_s d_{\pi_0}(s) \left[ A_{\pi_0}(s, \pi) \right]$$

measures whether the new policy takes actions with positive advantage at states visited by $\pi_0$. This term is linear in $\pi$. The remainder term in this Taylor expansion

$$\text{Rem}_{\pi_0}(\pi) = \sum_s \left( d_\pi(s) - d_{\pi_0}(s) \right) A_{\pi_0}(s, \pi)$$

captures distribution shift — changes in the aggregate advantage under the occupancy $d_\pi$ relative to $d_{\pi_0}$.

## 4.2. A Generic Template for Trust Region Methods

Trust region methods maximize the local approximation to policy performance $L_{\pi_0}(\pi)$ while constraining how far the new policy can deviate from the old, so that local approximation remains accurate. They usually take the form:

$$\max_\pi L_{\pi_0}(\pi) \text{ s.t. } \sum_s w(s) D(\pi(\cdot|s)\,; \pi_0(\cdot|s)) \leq \Gamma. \quad (1)$$

Instantiating this template typically requires three key objects: (1) *the state importance-weights* $w(\cdot)$, (ii) *the divergence measure* $D$ which defines the sensitivity to possible deviations from the behavior policies action probabilities and (iii) *the radius* $\Gamma$, which must delicately balance improvement potential and safety. A rich line of theoretical work studies the convergence of local policy improvement methods (Kakade & Langford, 2002; Agarwal et al., 2021; Bhandari & Russo, 2024; Cen et al., 2022).

## 4.3. The Optimization Problem Solved by Success Conditioning

The next theorem shows that success conditioning is the *exact* solution to the following trust region problem:

$$\max_\pi L_{\pi_0}(\pi)$$
$$\text{s.t. } \sum_s d_{\pi_0}^+(s) \chi^2(\pi(\cdot|s)\|\pi_0(\cdot|s)) \leq \sum_s d_{\pi_0}^+(s) \mathcal{I}_{\pi_0}(s)$$

In words, success conditioning exactly maximizes the first-order policy optimization $L_{\pi_0}(\pi)$ subject to the constraint that the policy moves no more in $\chi^2$ divergence than the aggregate action-influence under the behavior policy. Here $d_{\pi_0}^+$ denotes the state-occupancy under *successful* trajectories generated by the behavior policy (See Sec. 2.1).

**Proposition 4.1.** *The success-conditioned policy $\pi_+$ is an optimal solution to the trust region problem* (1) *with*

1. *State importance weights equal to the success-conditioned occupancy measure:* $w(s) = d_{\pi_0}^+(s)$.

2. *Chi-squared divergence:* $D = \chi^2(\cdot\|\cdot)$.

3. *Radius* $\Gamma = \sum_s d_{\pi_0}^+(s) \mathcal{I}_{\pi_0}(s)$.

The $\chi^2$ divergence is defined below and compared carefully to KL divergence measures in Section 4.5. Roughly, we will show this is a conservative measure, which heavily *penalizes exploring actions* that are extremely rare under $\pi_0$, but is *tolerant of dropping actions* the behavior policy explored.

**Definition 4.2.** For pmfs $P$ and $Q$ over an alphabet $\mathcal{X}$,

$$\chi^2(P\|Q) = \sum_{x \in \mathcal{X}} \left( \frac{P(x)}{Q(x)} - 1 \right)^2 Q(x) = \text{Var}_{x \sim Q} \left( \frac{P(x)}{Q(x)} \right).$$

## 4.4. An Identity Quantifying the Magnitude of Policy Improvement

The next lemma provides further insight into the optimizer $\pi_+$. At the optimum, the constraint is binding, so that the magnitude of policy change in $\chi^2$ divergence is exactly equal to aggregate action-influence. Moreover, both quantities are equal to the normalized objective value attained, $L_{\pi_0}(\pi_+)/\rho(\pi_0)$ — which quantifies the first-order policy improvement relative to the success rate of the base policy.

**Proposition 4.3** (Weighted Action-Influence-Identity). *Under the success-conditioned policy $\pi_+$,*

$$\underbrace{L_{\pi_0}(\pi_+)/\rho(\pi_0)}_{\textit{First-order improvement}} = \sum_s \underbrace{d_{\pi_0}^+(s)\chi^2(\pi_+(\cdot|s)\|\pi_0(\cdot|s))}_{\textit{Magnitude of policy change}}$$

$$= \sum_s \underbrace{d_{\pi_0}^+(s)\mathcal{I}_{\pi_0}(s)}_{\textit{Action-influence}}$$

This remarkable property indicates that success conditioning is always conservative — it never moves more than it can improve — and its failure mode arises when action-influence is very small, in which case it stays at the behavior policy and does little of consequence, just as in Example 3.1.

Among these three equal quantities, **the magnitude of policy change is particularly useful as a diagnostic**. Policy improvement is the object of interest, but estimating $\rho(\pi^+) - \rho(\pi_0)$ directly requires deploying $\pi^+$ to collect fresh rollouts *and* labeling their outcomes, which may be expensive. Action-influence determines the achievable magnitude of improvement but depends on value functions that success conditioning avoids estimating. By contrast, the $\chi^2$ divergence between $\pi_+$ and $\pi_0$ is computable from the two policies alone, with no new rollouts and no new reward labels: it requires only querying both models at states visited along the already-labeled successful trajectories used for training. (See A) Proposition 4.3 links these quantities exactly, making policy movement an ex ante diagnostic of both improvement and action-influence.

**Beyond the first-order objective.** Proposition 4.3 is a corollary of the following more refined equality that holds at every state individually.

**Proposition 4.4.** *For any state $s$,*

$$\underbrace{\frac{[A_{\pi_0}(s,\pi_+)]}{V_{\pi_0}(s)}}_{\textit{Relative advantage}} = \underbrace{\chi^2\left(\pi_+(s,\cdot)\|\pi_0(s,\cdot)\right)}_{\textit{Magnitude of policy change}} = \underbrace{\mathcal{I}_{\pi_0}(s)}_{\textit{Action-influence}}.$$

This more refined result lets us show that success conditioning improves the true success probabilities ($\rho(\pi)$ and/or $V_\pi(s)$) and not just the first-order approximation. A formula for the exact improvement $\rho(\pi_+) - \rho(\pi_0)$ can be given using the performance difference lemma (see Appendix C).

**Corollary 4.5.** $\rho(\pi_+) \geq \rho(\pi_0)$ *and, at any state $s$,*

$$\frac{V_{\pi_+}(s) - V_{\pi_0}(s)}{V_{\pi_0}(s)} \geq \mathcal{I}_{\pi_0}(s) \geq 0.$$

### 4.5. Comparison with Other Trust Region Optimization Problems

While success conditioning appears unrelated to explicit policy optimization—requiring only supervised learning

on filtered trajectories—we have shown that the Bayesian update defining $\pi_+$ is equivalently the solution to a trust-region optimization problem. This trust-region formulation differs in subtle but important ways from those commonly studied in the literature, which we now discuss.

The broad view that emerges is that success conditioning is a *cautious* method. This is reflected in the geometry of its trust region: the $\chi^2$ constraint heavily penalizes assigning significant probability to actions that were rare under the behavior policy. It is also reflected in the automatically determined trust-region radius, which prevents degradation in policy performance but may severely limit improvement when action-influence is small. We now compare this optimization problem to the KL-constrained formulations that dominate the literature.

#### 4.5.1. COMPARISON WITH REVERSE KL (TRPO).

Trust Region Policy Optimization (TRPO) (Schulman et al., 2015) and its widely used approximation, Proximal Policy Optimization (PPO) (Schulman et al., 2017) are among the most widely used RL algorithms. TRPO aims to solve

$$\texttt{TRPO:} \quad \max_\pi \; L_{\pi_0}(\pi)$$
$$\text{s.t.} \quad \sum_s d_{\pi_0}(s)\, D_{\text{KL}}\big(\pi_0(\cdot \mid s)\,\|\,\pi(\cdot \mid s)\big) \leq \Gamma.$$

This 'reverse' KL constraint penalizes the new policy for *dropping* actions taken by the old policy, enforcing coverage; any feasible policy must remain stochastic whenever the behavior policy was. By contrast, the trust-region problem corresponding to success conditioning replaces the reverse KL divergence with a $\chi^2$ constraint weighted by the success-conditioned occupancy measure. This change reverses the geometric bias of the trust region: *instead of penalizing the removal of previously taken actions, the $\chi^2$ constraint penalizes assigning probability to actions that were rarely taken under the behavior policy.* Consequently, a policy satisfying a $\chi^2$ constraint may concentrate its probability mass on a single action, provided that action was well supported under the behavior policy (Figure 1).

#### 4.5.2. COMPARISON WITH FORWARD KL (MDPO).

Alternatives to TRPO based on forward KL or entropy-regularized objectives have also been studied, including regularized Markov decision processes (Geist et al., 2019) and mirror-descent policy optimization (MDPO)(Tomar et al., 2022). MDPO solves

$$\texttt{MDPO:} \quad \max_\pi \; L_{\pi_0}(\pi)$$
$$\text{s.t.} \quad \sum_s d_{\pi_0}(s)\, D_{\text{KL}}\big(\pi(\cdot \mid s)\,\|\,\pi_0(\cdot \mid s)\big) \leq \Gamma.$$

This 'forward' KL constraint is qualitatively closer to the $\chi^2$ constraint: both penalize deviations outside the support

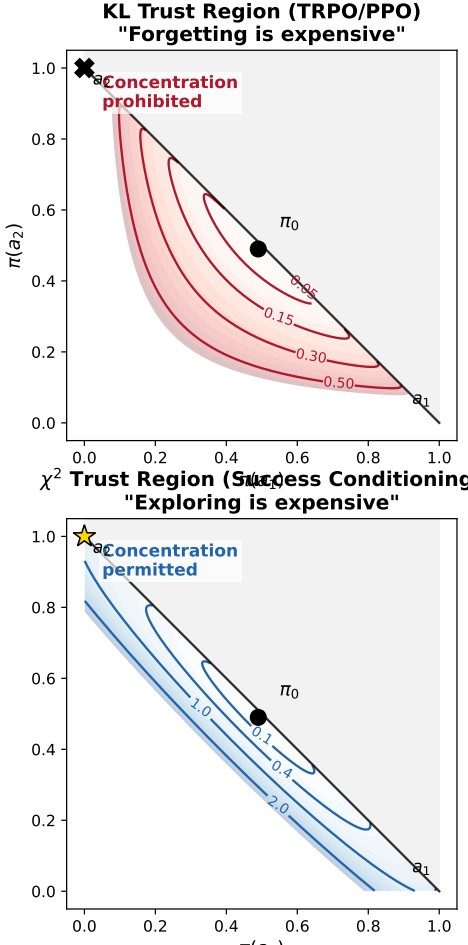

*Figure 1.* Trust region geometry for a three-action problem with $\pi_0 = (0.49, 0.49, 0.02)$. The simplex constraint restricts policies to the region below the diagonal. **Top:** KL prohibits concentration. **Bottom:** $\chi^2$ permits concentration but penalizes shifting mass toward actions that are rare under $\pi_0$.

of the behavior policy. However, the $\chi^2$ constraint is substantially more restrictive. Lemma 4.6 makes this precise: under a fixed forward KL budget, the maximum probability assignable to a rare action of the behavior policy with probability $\delta$ scales as $1/\log(1/\delta)$, whereas under a $\chi^2$ constraint it scales as $\sqrt{\delta}$. For rare actions, $\sqrt{\delta} \ll 1/\log(1/\delta)$, so the $\chi^2$ trust region keeps the policy much closer to observed behavior.

**Lemma 4.6** (Tolerance for exploring rare actions)**.** *Fix an integer $k \geq 2$. Consider a single state with action set $\{a^\star, a_1, \ldots, a_k\}$ and behavior policy*

$$\pi_0(a^\star) = \delta, \qquad \pi_0(a_i) = \frac{1-\delta}{k} \quad (i = 1, \ldots, k),$$

*where $\delta \in (0, 1)$ is the probability of the rare action $a^\star$. For any candidate policy $\pi$, write $p := \pi(a^\star)$.*

**Algorithm 1** Iterated Success Conditioning (population)

---

**Input:** parameters $\theta$, inner steps $N$, step size $\eta$
**repeat**
  $\theta_0 \leftarrow \theta; \quad \pi_0 \leftarrow \pi_{\theta_0}$        {freeze behavior policy}
  **for** $i = 1$ **to** $N$ **do**
    $\theta \leftarrow \theta - \eta \, \nabla_\theta \, \ell_{\pi_0}(\pi_\theta)$
  **end for**
**until** converged

---

*Fix a constant divergence budget $\varepsilon > 0$ independent of $\delta$. Then,*

$$\sup \left\{ \pi(a^\star) : \ \chi^2(\pi \| \pi_0) \leq \varepsilon \right\} = \Theta(\sqrt{\delta}),$$

*whereas,*

$$\sup \left\{ \pi(a^\star) : \ D_{\mathrm{KL}}(\pi \| \pi_0) \leq \varepsilon \right\} = \Theta\left(\frac{1}{\log(1/\delta)}\right).$$

*The implied constants depend on $\varepsilon$ but not on $k$.*

## 5. Relationship to Policy Gradient Methods

Success conditioning is defined by an imitation loss: it returns the target policy $\pi$ minimizing the next-action prediction loss along successful trajectories under the behavior policy $\pi_0$,

$$\ell_{\pi_0}(\pi) = \mathbb{E}_{\tau \sim P_{\pi_0}(\cdot | R=1)}\left[ -\sum_{t=1}^{T-1} \log \pi(A_t \mid S_t) \right].$$

Section 4 recharacterizes its minimizer $\pi_+$ as the solution of a coherent policy optimization problem. Here we relate the procedure that minimizes this loss to standard policy gradient methods, finding that REINFORCE emerges as a special case in which the behavior policy is refreshed after every update.

Connecting to gradient methods requires, for the first time, a parameterized policy $\pi_\theta$; we state both procedures as population idealizations, and note their finite-sample form in the prose. Algorithm 1 is iterated success conditioning. In an outer loop it freezes a behavior policy $\pi_0 := \pi_{\theta_0}$ and forms the imitation loss $\ell_{\pi_0}$; in an inner loop it takes $N$ gradient steps on that fixed loss. In practice the expectation defining $\ell_{\pi_0}$ is approximated by sampling trajectories from $\pi_0$, retaining those with $R = 1$, and training on the resulting successful dataset. A salient aspect of the procedure is that it is **off-policy** for $N > 1$: across the inner steps the target policy $\pi_\theta$ drifts away from the behavior policy $\pi_0$ whose data it is trained on.

**REINFORCE as On-Policy Incremental Success Conditioning.** What happens if instead we set $N = 1$, so that the target policy is always updated with fresh on-policy trajectories? It turns out that this implements the policy gradient

---

**Algorithm 2** REINFORCE, binary reward (population)

---

**Input:** parameters $\theta$, step size $\eta$
**repeat**
    $\theta_0 \leftarrow \theta; \quad \pi_0 \leftarrow \pi_{\theta_0}$
    $\theta \leftarrow \theta + \eta \nabla_\theta \, \rho(\pi_\theta)\big|_{\theta=\theta_0}$     {policy gradient on $\rho$}
**until** converged

---

algorithm REINFORCE (Alg 2), up to a step-size rescaling. This is the content of the following, simple, identity, where $\rho(\pi_\theta) = \mathbb{E}_{\pi_\theta}[R(\tau)]$.

**Lemma 5.1.** *For any $\theta_0$ with behavior policy $\pi_0 = \pi_{\theta_0}$,*

$$-\nabla_\theta \, \ell_{\pi_0}(\pi_\theta)\big|_{\theta=\theta_0} \;=\; \frac{1}{\rho(\pi_0)} \nabla_\theta \, \rho(\pi_\theta)\big|_{\theta=\theta_0}.$$

The identity follows by differentiating $\ell_{\pi_0}$, using that conditioning on the binary event $R = 1$ reweights each trajectory by $R(\tau)/\rho(\pi_0)$, together with the score-function identity $\nabla_\theta \rho(\pi_\theta) = \mathbb{E}_{\pi_\theta}[R(\tau) \sum_t \nabla_\theta \log \pi_\theta(A_t \mid S_t)]$. Lemma 5.1 shows that, at $\theta = \theta_0$, the gradient of the success-conditioning loss coincides with the REINFORCE direction, up to the positive scalar $1/\rho(\pi_0)$, which can be absorbed into the step size. If $N = 1$, every update of Algorithm 1 is taken at $\theta = \theta_0$ and aligns exactly with the REINFORCE gradient step. If $N > 1$, however, the inner loop continues descending the fixed loss $\ell_{\pi_0}$ after $\pi_\theta$ has moved away from $\pi_0$. These later steps are no longer policy-gradient steps for the current policy; they are supervised steps toward the success-conditioned target of the frozen $\pi_0$, whose minimizer is $\pi_+$.

This places policy gradient *inside* the success-conditioning framework, rather than reducing success conditioning to policy gradient: REINFORCE is the incremental, on-policy special case that refreshes the behavior policy after every step. The distinction is practically important because generating and labeling trajectories is often far more costly than additional supervised steps on data already in hand. If we insist that every update be a gradient step for the current policy, reusing older data becomes an off-policy problem requiring importance weighting or other corrections. Success conditioning offers a different primitive: given trajectories from a behavior policy, one defines the target obtained by conditioning that behavior on success, and fits it. The successful trajectories from the behavior policy are then not stale samples for a current-policy gradient, but precisely the samples defining the supervised target. This view also suggests that deliberately separating the data-generating policy from the target — for instance, by injecting exploration into data collection — is a promising direction.

## 6. Success Conditioning from Data

We have shown that the success-conditioned policy $\pi^+$ solves a trust-region optimization problem with guaranteed improvement. Since $\pi^+$ is equivalently the minimizer of the next-action prediction loss on successful trajectories (Section 2), a natural question is whether fitting this objective well offline translates to good deployment performance.

The answer takes a familiar form: deployment error is bounded by offline loss times a *distribution shift* term. The distribution shift is captured by the ratio

$$M := \sup_s \frac{d_{\pi_0}^+(s)}{d_{\pi_+}(s)}.$$

which has a clear semantic meaning: it compares occupancy under the learned policy to occupancy success-conditioned trajectories from $\pi_0$. (See Sec 2.1)

**Proposition 6.1.** *Suppose an estimated policy $\hat{\pi}$ achieves excess cross-entropy loss $\Delta = \ell(\hat{\pi}) - \ell(\pi_+)$ relative to the success-conditioned policy. Then its success probability satisfies $|\rho(\hat{\pi}) - \rho(\pi_+)| \leq \sqrt{M \cdot \Delta/2}$.*

Such bounds are ubiquitous in offline RL, where distribution-shift terms are often difficult to interpret or control. Here, $M = 1$ whenever state transitions are deterministic—including autoregressive sequence generation. By definition, success-conditioned trajectories have the same conditional action distribution as executing $\pi_+$, and under deterministic dynamics this implies the same induced state occupancy. In such settings, the supervised objective aligns directly with deployment performance, with no distribution-shift penalty.

## 7. Dense Rewards, Return Thresholding, and Influence Amplification

Many reinforcement learning problems involve dense rewards, but conditioning on success inherently requires specifying a binary success criterion. A common response is return thresholding: defining a trajectory as successful if the cumulative reward exceeds a specified threshold. Prior work has observed that conditioning on overly high thresholds can degrade average performance by "counting on luck," emphasizing stochastic outcomes rather than reliable action choices (Paster et al., 2022; Yang et al., 2023).

This section provides a precise perspective on this phenomenon. Section 7.1 defines a theoretically faithful reduction from dense rewards to binary ones, allowing our policy improvement guarantees to extend directly to this setting. Relative to this faithful binary reward, return thresholding can be viewed as a proxy success criterion—one that may differ from the objective used in evaluation. We extend our policy improvement analysis to such proxy rewards,

showing that they can amplify action-influence and hence improvement up to a point, but risk degrading performance when misalignment becomes severe.

## 7.1. Faithful Binary Rewards

We define a theoretically faithful reduction from dense rewards to binary ones, allowing our policy improvement guarantees to extend directly. Suppose the terminal return $Y = Y(\tau)$ takes values in $[0, 1]$ (or is normalized to this range). For analysis, we can equivalently view each trajectory as associated with a binary success variable $R(\tau) \in \{0, 1\}$ drawn once from Bernoulli($Y(\tau)$), so that $\mathbb{P}(R(\tau) = 1 \mid \tau) = Y(\tau)$. Once drawn, this label is fixed and treated as the trajectory's outcome.

This unbiased labeling preserves the original objective in expectation: since $\mathbb{E}[R(\tau)] = \mathbb{E}[Y(\tau)]$, maximizing success probability $\mathbb{P}(R(\tau) = 1)$ under any policy is equivalent to maximizing expected return $\mathbb{E}[Y(\tau)]$. Moreover, once labels are assigned, the setting matches our earlier framework exactly—$R$ is a deterministic function of the labeled trajectory, and all preceding results apply without modification. Success conditioning with respect to these binary labels therefore yields policy updates that improve the original expected-return objective.

## 7.2. Proxy Rewards and Their Effects

In practice, success is often defined using alternative, handcrafted criteria rather than a faithful reduction of the underlying reward. We refer to such choices as *proxy rewards*. A common example is return thresholding, where $\tilde{R}(\tau) = \mathbf{1}\{Y(\tau) > \theta\}$ (see Sec. 7.3). The next result provides an exact characterization of the benefits and risks of conditioning on proxy rewards.

Let $\tilde{R}$ be a proxy-reward. We use tildes to denote the corresponding objects, with $\tilde{Q}_{\pi_0}$ denoting the $Q$-function under the behavior policy, $\tilde{\mathcal{I}}_{\pi_0}(s)$ denoting the action-influence at state $s$ (the squared coefficient of variation of $\tilde{Q}_{\pi_0}$) and $\tilde{\pi}_+(s, a) = \mathbb{P}_{\pi_0}\left(A_t = a \mid S_t = s, \tilde{R}(\tau) = 1\right)$ denoting the corresponding success conditioned-policy.

**Proposition 7.1.** *For any non-terminal state $s$,*

$$\frac{A_{\pi_0}(s, \tilde{\pi}_+)}{A_{\pi_0}(s, \pi_+)} = \sqrt{\frac{\tilde{\mathcal{I}}_{\pi_0}(s)}{\mathcal{I}_{\pi_0}(s)}} \cdot \underbrace{\underset{a \sim \pi_0}{\text{Corr}}\left(A_{\pi_0}(s, a)\tilde{A}_{\pi_0}(s, a)\right)}_{alignment}.$$

Compared to faithful conditioning, proxy conditioning helps if the relative increase in action-influence (coefficient of variation of the $Q$ function) is strong enough to overcome any loss in alignment.

## 7.3. Insight into Return Thresholding

We use this framework to interpret return thresholding, where success is defined as $\tilde{R} = \mathbf{1}(Y > \theta)$. This approach is tightly linked to top-$k\%$ trajectory filtering and reminiscent of Decision Transformers (Chen et al., 2021), which condition on return equaling a specified value. In all these approaches, improvement on the true objective depends on both how discriminating the proxy is (its coefficient of variation) and how well it aligns with the true reward (their correlation)—exactly the tradeoff Prop 7.1 quantifies.

We illustrate this analysis using a stylized bandit example. There are 100 arms. Pulling arm $i$ yields reward $Y_i \sim \text{Beta}(a_i, b_i)$. Ninety-nine "moderate" arms have symmetric Beta distributions with mean $0.5$ but varying shapes ($a_i = b_i \sim \text{Uniform}(0.3, 0.7)$), yielding U-shaped distributions with mass at both extremes. One "special" arm has Beta($18, 2$), giving mean $0.9$ with mass concentrated near this value. The behavior policy is uniform.

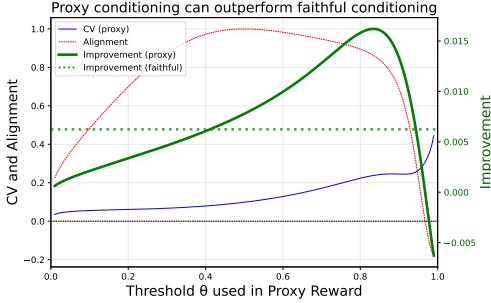

*Figure 2.* Alignment and improvement as a function of the success threshold used in the proxy reward.

This stylized example distills both the benefits and risks of return thresholding. As shown in Figure 2, there is a range of moderate-to-high thresholds where using the proxy reward increases performance on the true objective—substantially outperforming faithful conditioning. The proxy succeeds because it discriminates more sharply: the special arm reliably exceeds the threshold while the moderate arms mostly do not. Beyond a point, however, the method starts selecting for arms whose expected performance is poor but whose outcome distribution is volatile, causing occasional large outcomes that cross the threshold. When this happens, alignment degrades and improvement turns negative.

## 8. Related Literature

### 8.1. Return Conditioning in RL

A growing body of work studies *return- or success conditioned* policies, in which actions are generated conditional on a desired outcome—such as a target return or task completion—rather than via explicit policy optimization. Early

examples include Upside-Down Reinforcement Learning (Schmidhuber, 2019; Srivastava et al., 2019). This perspective was popularized by Decision Transformers (Chen et al., 2021) and related sequence-modeling approaches to offline RL (Janner et al., 2021), and later extended to alternative generative models (Ajay et al., 2023) and to more general notions of success and goals, as in Reinforcement via Supervised Learning (RvS) (Emmons et al., 2022).

Empirically, return-conditioned methods initially showed strong performance despite relying only on supervised learning, but subsequent work identified important limitations: optimal policy recovery requires strong assumptions (Brandfonbrener et al., 2022), and conditioning on very high returns may "count on luck," emphasizing stochastic outcomes over reliable choices (Paster et al., 2022; Yang et al., 2023). We show these failures arise from misalignment between the success criterion and the evaluation objective (Sec. 7); when aligned, success conditioning is a conservative improvement operator whose failure mode is over-conservatism when action-influence is small. This characterization is not provided by prior analyses; the closest is work of (Eysenbach et al., 2022), which notes the form of the success-conditioned policy and its improvement in single-goal settings, but does not characterize it as a policy improvement operator or analyze its failure modes.

### 8.2. Exponential Policy Updates and Imitation Learning

A large fraction of reinforcement learning algorithms update policies by *exponentially reweighting* actions according to observed or estimated outcomes (e.g., returns, advantages, or Q-values) or preference-based comparisons, as in Direct Preference Optimization (Rafailov et al., 2023). The resulting Gibbs policies are equivalent to KL- or entropy-regularized updates and arise across control-as-inference and variational formulations (Peters & Schaal, 2007), information-theoretic optimization (Peters et al., 2010; Haarnoja et al., 2018), EM-style methods (Abdolmaleki et al., 2018), and conservative offline RL (Peng et al., 2020; Wang et al., 2020; Kostrikov et al., 2022). Despite their diverse motivations, these methods share a common structure: scalar quality weights tilt the behavior distribution toward higher-valued actions.

A similarity with success conditioning is that these methods admit an *imitation-learning interpretation*. Policies of the form $\pi(a \mid s) \propto \exp(\beta Q(s, a))$ arise as solutions to the weighted maximum-likelihood problem

$$\max_{\pi} \; \mathbb{E}_{(s,a) \sim D}[\exp(\beta Q(s, a)) \log \pi(a \mid s)].$$

This objective has the same form as behavior cloning, but with the empirical distribution *tilted* toward higher-valued actions (Peters & Schaal, 2007; Peng et al., 2020; Kostrikov et al., 2022). Success conditioning shares this supervised-

learning form but differs in construction: its weights arise from exact conditioning on a terminal success event, involve no tunable temperature parameter $\beta$, and we show it induces a $\chi^2$ rather than KL geometry.

### 8.3. Control as Inference

Control-as-inference formulations introduce auxiliary "optimality" variables and perform Bayesian conditioning in an augmented graphical model. We refer to the tutorial and review of (Levine, 2018). In the exact formulation, conditioning on optimality yields a posterior over trajectories proportional to the dynamics probability times an exponential of accumulated reward. (Levine, 2018, Section 2) This construction is closely related in spirit to success conditioning, but corresponds to exponential tilting of trajectories rather than conditioning on a hard terminal success event.

To derive practical algorithms in stochastic environments, control-as-inference departs more substantially from success conditioning (Levine, 2018, Section 3). Through variational approximations, the inference problem is converted into a KL-regularized control problem—effectively a regularized MDP—by fixing the dynamics and optimizing only the policy. The resulting objective is then addressed using dynamic programming or iterative policy improvement algorithms such as Soft Actor-Critic (Haarnoja et al., 2018). In contrast, success conditioning does not define a new control problem to be solved by iterative policy search; it specifies a single policy update directly via Bayes' rule, conditioning the behavior policy on observed success.

## 9. Limitations and Open Directions

Our analysis characterizes exact success conditioning under a fixed behavior policy, clarifying the target policy that common training procedures aim to approximate. While the results consist of exact equalities and do not rely on asymptotic arguments or empirical validation, they do not—except for Proposition 6.1—address the effects of finite data, function approximation, or large-scale optimization, which we view as a distinct question that is important but likely sensitive to specific algorithmic choices.

An important open direction is to understand how the policy movement metric (Sec. 4.4) behaves in realistic training regimes, and when it serves as a useful diagnostic of improvement during large-scale success conditioning. On the theoretical side, further work could analyze iterated success conditioning and its convergence properties. Our results already imply that each iteration (weakly) improves the policy. The challenge is that success conditioning has no forced exploration, and progress could stall if it collapses on a degenerate policy. Future work could also look at mechanisms such as action chunking that may increase action-influence.

## Impact statement

This paper presents work whose goal is to advance the field of Machine Learning. There are many potential societal consequences of our work, none which we feel must be specifically highlighted here.

## Acknowledgments

Numerous conversions since submitting to ICML have greatly sharpened my understanding. Special thanks especially to the anonymous reviewers, Ian Osband, Ofir Nachum, Yuda Song, and Drew Bagnell

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

## A. Estimating the Policy-Movement Diagnostic

By Proposition 4.3, the first-order relative improvement equals the aggregate policy movement

$$\Gamma := \sum_s d_{\pi_0}^+(s)\,\delta(s), \qquad \delta(s) := \chi^2\big(\pi_+(\cdot|s) \,\|\, \pi_0(\cdot|s)\big).$$

We observe that $\Gamma$ is the expectation of a simple per-trajectory statistic, and hence is estimated by a sample average.

The success-conditioned occupancy is the expected state-visitation count along successful trajectories, $d_{\pi_0}^+(s) = \mathbb{E}\big[\sum_t \mathbf{1}(S_t = s)\big]$, where throughout this section the expectation is over $\tau \sim P_{\pi_0}(\cdot \mid R(\tau) = 1)$. Since $\delta(s)$ depends only on the state,

$$\Gamma = \mathbb{E}\left[\sum_{t=1}^{T-1} \delta(S_t)\right].$$

The per-state term is computed directly from the two policies,

$$\delta(s) = \sum_a \frac{\big(\pi_+(a|s) - \pi_0(a|s)\big)^2}{\pi_0(a|s)}.$$

Given a held-out set of $N$ successful trajectories $\{\tau^{(i)}\}_{i=1}^N$ from $\pi_0$, the estimator

$$\widehat{\Gamma} = \frac{1}{N} \sum_{i=1}^N \sum_t \delta\big(S_t^{(i)}\big)$$

is an average of i.i.d. per-trajectory terms, and is therefore an unbiased and consistent estimate of $\Gamma$. Each term requires only the next-action distributions of $\pi_+$ and $\pi_0$ at the visited states, obtained from one forward pass per model. Note that $\delta(s)$ must be evaluated using the full action distributions: replacing it with the single realized action $A_t$ is biased, since along successful trajectories $A_t \sim \pi_+(\cdot|S_t)$ rather than $\pi_0(\cdot|S_t)$.

## B. Proofs

### B.1. Expression for the Success-Conditioned Policy

Here we state a formula for the success-conditioned policy.

**Lemma B.1.** *For any state $s$ and action $a$,*

$$\pi_+(s,a) = \pi_0(a|s)\left(1 + \frac{A_{\pi_0}(s,a)}{V_{\pi_0}(s)}\right)$$

*Proof.* Applying Bayes rule, and the definitions in Section 2.1 yields

$$
\begin{aligned}
&\pi_+(s,a) \\
&= \mathbb{P}_{\pi_0}\left(A_t = a \mid S_t = s, R(\tau) = 1\right) \\
&= \frac{\mathbb{P}_{\pi_0}\left(A_t = a \wedge R(\tau) = 1 \mid S_t = s\right)}{\mathbb{P}_{\pi_0}\left(R(\tau) = 1 \mid S_t = s\right))} \\
&= \frac{\mathbb{P}_{\pi_0}\left(A_t = a \mid S_t = s\right)\mathbb{P}_{\pi_0}\left(R(\tau) = 1 \mid S_t = s, A_t = a\right)}{\mathbb{P}_{\pi_0}\left(R(\tau) = 1 \mid S_t = s\right)} \\
&= \frac{\pi_0(a|s)Q_{\pi_0}(s,a)}{V_{\pi_0}(s)} \\
&= \pi_0(a|s)\left(1 + \frac{A_{\pi_0}(s,a)}{V_{\pi_0}(s)}\right),
\end{aligned}
$$

where the final step uses that $Q_{\pi_0}(s,a) = V_{\pi_0}(s) + A_{\pi_0}(s,a)$. $\square$

## B.2. Expression for the Success-Conditioned Occupancy Measure

**Lemma B.2.** *The success-conditioned occupancy measure is related to the unconditioned occupancy measure as*

$$d_\pi^+(s) = \left(\frac{V_\pi(s)}{\rho(\pi)}\right) \cdot d_\pi(s)$$

*for any policy $\pi$ and state $s$.*

*Proof.*

$$
\begin{aligned}
\rho(\pi) \cdot d_\pi^+(s) &= \rho(\pi)\mathbb{E}_\pi\left[\sum_{t=1}^{T-1}\mathbb{1}(S_t = s) \mid R(\tau) = 1\right] \\
&= \mathbb{E}_\pi\left[\sum_{t=1}^{T-1}\mathbb{1}(S_t = s) \cdot \mathbb{1}(R(\tau) = 1)\right] \\
&= \mathbb{E}_\pi\left[\sum_{t=1}^{T-1}\mathbb{E}_\pi\left[\mathbb{1}(S_t = s) \cdot \mathbb{1}(R(\tau) = 1) \mid S_t\right]\right] \\
&= \mathbb{E}_\pi\left[\sum_{t=1}^{T-1}\mathbb{1}(S_t = s) \cdot \mathbb{P}(R(\tau) = 1 \mid S_t)\right] \\
&= \mathbb{E}_\pi\left[\sum_{t=1}^{T-1}\mathbb{1}(S_t = s) \cdot V_\pi(S_t)\right] \\
&= \mathbb{E}_\pi\left[\sum_{t=1}^{T-1}\mathbb{1}(S_t = s) \cdot V_\pi(s)\right] \\
&= V_\pi(s)d_\pi(s).
\end{aligned}
$$

$\square$

## B.3. Proof of Proposition 4.1

**Lemma B.3.** *Define the normalized state distribution*

$$h(s) := \frac{d_{\pi_0}^+(s)}{\sum_{s'} d_{\pi_0}^+(s')}. \tag{2}$$

*and let $(S, A)$ be sampled according to*

$$S \sim h, \qquad A \mid S \sim \pi_0(\cdot|s). \tag{3}$$

*Then, the success conditioned optimization problem, is equivalent to*

$$\max_{\pi \in \Pi} \mathbb{E}\left[\left(\frac{\pi(A \mid S)}{\pi_0(A \mid S)} - 1\right) \times \frac{A_{\pi_0}(S,A)}{V_{\pi_0}(S)}\right] \tag{4}$$

$$s.t \; \mathbb{E}\left[\left(\frac{\pi(A \mid S)}{\pi_0(A \mid S)} - 1\right)^2\right] \leq \mathbb{E}\left[\left(\frac{A_{\pi_0}(S,A)}{V_{\pi_0}(S)}\right)^2\right] \tag{5}$$

*where $\Pi$ denotes the set of all stochastic policies.*

*Proof.* By Lemma B.2, the linearized improvement objective can be written as

$$L_{\pi_0}(\pi) = \rho(\pi_0)\sum_s d_{\pi_0}^+(s)\,\mathbb{E}_{a\sim\pi(\cdot|s)}\left[\frac{A_{\pi_0}(s,a)}{V_{\pi_0}(s)}\right].$$

Using a change of measure and dropping constants that depend on $\pi_0$ only, this becomes

$$L_{\pi_0}(\pi) \propto \mathbb{E}\left[\frac{\pi(A \mid S)}{\pi_0(A \mid S)} \cdot \frac{A_{\pi_0}(S,A)}{V_{\pi_0}(S)}\right]. \tag{6}$$

By the definition of the advantage, $\mathbb{E}\left[A_{\pi_0}(S,A) \mid S\right] = 0$, so we can equivalently write the objective as

$$L_{\pi_0}(\pi) \propto \mathbb{E}\left[\left(\frac{\pi(A \mid S)}{\pi_0(A \mid S)} - 1\right) \cdot \frac{A_{\pi_0}(S,A)}{V_{\pi_0}(S)}\right]. \tag{7}$$

We now rewrite the trust-region constraint. Using the standard identity

$$\chi^2(P\|Q) = \mathbb{E}_{x\sim Q}\left[\left(\frac{P(x)}{Q(x)} - 1\right)^2\right], \tag{8}$$

the constraint

$$\sum_s d_{\pi_0}^+(s)\,\chi^2(\pi(\cdot \mid s) \| \pi_0(\cdot \mid s)) \leq \sum_s d_{\pi_0}^+(s)\,\mathcal{I}_{\pi_0}(s) \tag{9}$$

is equivalent (up to the same normalization constant) to

$$\mathbb{E}\left[\left(\frac{\pi(A \mid S)}{\pi_0(A \mid S)} - 1\right)^2\right] \leq \mathbb{E}\left[\left(\frac{A_{\pi_0}(S,A)}{V_{\pi_0}(S)}\right)^2\right].$$

To rewrite the expected action-advantage, $\mathbb{E}\left[\mathcal{I}_{\pi_0}(S)\right]$, we applied the definition $\mathcal{I}_{\pi_0}(S) = \mathbb{E}\left[\frac{A_{\pi_0}(S,A)}{V_{\pi_0}(S)} \mid S\right]$ and the law of iterated expectations. $\square$

*Proof of Proposition 4.1.* Fix $\pi_0$. Let $h$ and $(S, A)$ be sampled as in Lemma B.3:

For any feasible $\pi$, Cauchy–Schwarz gives

$$\mathbb{E}\left[\left(\frac{\pi(A \mid S)}{\pi_0(A \mid S)} - 1\right)\frac{A_{\pi_0}(S, A)}{V_{\pi_0}(S)}\right]$$

$$\leq \sqrt{\mathbb{E}\left[\left(\frac{\pi(A \mid S)}{\pi_0(A \mid S)} - 1\right)^2\right]} \times \sqrt{\mathbb{E}\left[\left(\frac{A_{\pi_0}(S, A)}{V_{\pi_0}(S)}\right)^2\right]}$$

$$\leq \mathbb{E}\left[\left(\frac{A_{\pi_0}(S, A)}{V_{\pi_0}(S)}\right)^2\right]. \tag{10}$$

By Lemma B.1,

$$\pi_+(a \mid s) = \pi_0(a \mid s)\left(1 + \frac{A_{\pi_0}(s, a)}{V_{\pi_0}(s)}\right),$$

which implies

$$\frac{\pi_+(A \mid S)}{\pi_0(A \mid S)} - 1 = \frac{A_{\pi_0}(S, A)}{V_{\pi_0}(S)}. \tag{11}$$

Substituting (11) into (9) shows the rewritten feasibility constraint (5) holds with equality. Substituting into (4) shows $\pi_+$ achieves the upper bound in (10). Therefore $\pi_+$ is optimal for (4)–(5), and hence for the original trust-region problem in Proposition 4.1. $\qquad\square$

## B.4. Proof of Proposition 4.4

*Proof.* Using the formula for the success-conditioned policy $\pi_+(a|s)$ in Lemma B.1 yields:

$$\mathbb{E}_{a \sim \pi_+(\cdot|s)}\left[A_{\pi_0}(s, a)\right]$$

$$= \sum_a \pi_+(a|s)A_{\pi_0}(s, a)$$

$$= \sum_a \left[\pi_0(a|s)\left(1 + \frac{A_{\pi_0}(s, a)}{V_{\pi_0}(s)}\right)\right]A_{\pi_0}(s, a)$$

$$= \sum_a \pi_0(a|s)A_{\pi_0}(s, a) + \frac{\sum_a \pi_0(a|s)\left(A_{\pi_0}(s, a)\right)^2}{V_{\pi_0}(s)}.$$

$$= 0 + \frac{\sum_a \pi_0(a|s)\left(A_{\pi_0}(s, a)\right)^2}{V_{\pi_0}(s)},$$

where the final equality uses that the average advantage of actions drawn from $\pi_0$ over $\pi_0$ is zero. Dividing by $V_{\pi_0}(s)$ yields,

$$\frac{\mathbb{E}_{a \sim \pi_+(\cdot|s)}\left[A_{\pi_0}(s, a)\right]}{V_{\pi_0}(s)} = \frac{\sum_a \pi_0(a|s)\left(A_{\pi_0}(s, a)\right)^2}{\left(V_{\pi_0}(s)\right)^2}$$

$$= \sum_a \pi_0(a|s)\left(\frac{A_{\pi_0}(s, a)}{V_{\pi_0}(s)}\right)^2$$

$$= \mathcal{I}_{\pi_0}(s).$$

Now we show

$$\chi^2\left(\pi_+(\cdot|s)||\pi_0(\cdot|s)\right) = \mathcal{I}_{\pi_0}(s) \tag{12}$$

Using Definition 4.2,

$$\chi^2\left(\pi_+(\cdot|s)||\pi_0(\cdot|s)\right) = \mathrm{Var}_{a \sim \pi_0(\cdot|s)}\left(\frac{\pi_+(a|s)}{\pi_0(a|s)}\right).$$

$$= \mathrm{Var}_{a \sim \pi_0(\cdot|s)}\left(1 + \frac{A_{\pi_0}(s, a)}{V_{\pi_0}(s)}\right)$$

$$= \mathrm{Var}_{a \sim \pi_0(\cdot|s)}\left(\frac{A_{\pi_0}(s, a)}{V_{\pi_0}(s)}\right)$$

$$= \mathbb{E}_{a \sim \pi_0(\cdot|s)}\left(\frac{A_{\pi_0}(s, a)}{V_{\pi_0}(s)}\right)^2.$$

The last step uses that for any random variable $X$, $\mathbb{E}[X^2] = \mathrm{Var}(X) + \left(\mathbb{E}[X]\right)^2$. In our case, the random variable ($X$) has mean zero, since $\mathbb{E}_{a \sim \pi_0(\cdot|s)}\left[A_{\pi_0}(s, a)\right] = \mathbb{E}_{a \sim \pi_0(\cdot|s)}\left[Q_{\pi_0}(s, a)\right] - V_{\pi_0}(s) = 0$ $\qquad\square$

## B.5. Proof of Proposition 4.3

*Proof.* We begin by recalling the definition

$$A_{\pi_0}(s, \pi_+) := \mathbb{E}_{a \sim \pi_+(\cdot|s)}\left[A_{\pi_0}(s, a)\right].$$

By Proposition 4.4, we have

$$\frac{A_{\pi_0}(s, \pi_+)}{V_{\pi_0}(s)} = \mathcal{I}_{\pi_0}(s) = \chi^2(\pi_+(\cdot \mid s) \| \pi_0(\cdot \mid s)),$$

and hence

$$A_{\pi_0}(s, \pi_+) = V_{\pi_0}(s)\,\mathcal{I}_{\pi_0}(s).$$

Multiplying by the (unconditioned) occupancy measure $d_{\pi_0}(s)$ and summing over $s$ yields

$$L_{\pi_0}(\pi_+) = \sum_s d_{\pi_0}(s)V_{\pi_0}(s)\mathcal{I}_{\pi_0}(s)$$

$$= \rho(\pi_0)\sum_s d_{\pi_0}^+(s)\mathcal{I}_{\pi_0}(s).$$

The first equality is a definition of $L_{\pi_0}(\pi_+)$. The second is the formula for the success-conditioned occupancy measure in Lemma B.2. Dividing each side by the success-probability under the behavior policy, $\rho(\pi_0)$, yields the result. We've focused in the $\mathcal{I}_{\pi_0}$ expression, since an identical argument establishes the claim for the $\chi^2$ expression. $\qquad\square$

## B.6. Proof of Corollary 4.5

This proof uses standard dynamic programming theory and is provided for completeness.

*Proof.* Let $T_\pi$ be the Bellman operator with respect to a policy $\pi$. This operator maps a value function $V$ to a new

value function $T_\pi V$. For a value function which is itself the expected value-to-go from another policy $\pi'$, it can be written as

$$(T_\pi V_{\pi'})(s) = \sum_a \pi(a|s) Q_{\pi'}(s, a) = V_{\pi'}(s) + A_{\pi'}(s, \pi).$$

Hence, our theory has shown that

$$\left(T_{\pi_+} V_{\pi_0}\right)(s) \geq V_{\pi_0}(s) \quad \forall s.$$

We write this element-wise inequality as

$$T_{\pi_+} V_{\pi_0} \succeq V_{\pi_0}.$$

The monotonicity property of the Bellman operator (Bertsekas, 2011) implies that this inequality is preserved under application of $T_\pi$, so $T_{\pi_+}^{(2)} V_{\pi_0} \succeq T_{\pi_+} V_{\pi_0}$. Repeating this inductively yields

$$V_{\pi_0} \preceq T_{\pi_+} V_{\pi_0} \preceq \cdots \preceq T_{\pi_+}^{(k)} V_{\pi_0} \to V_{\pi_+}.$$

Therefore, we've shown

$$V_{\pi_+}(s) \geq T_{\pi_+} V_{\pi_0}(s) = V_{\pi_0}(s) + A_{\pi_0}(s, \pi_+)$$

and our result follows. The claim that $\rho(\pi_+) \geq \rho(\pi_0)$ is immediate since $\rho(\pi) = \sum_s \mu(s) V_\pi(s)$ for any policy $\pi$. (Recall $\mu$ is the initial state distribution).

$\square$

### B.7. Proof of Lemma 4.6

*Proof.* Define

$$p_{\chi^2}^\star(\delta) := \sup\left\{\pi(a^\star): \ \chi^2(\pi\|\pi_0) \leq \varepsilon\right\},$$

$$p_{\mathrm{KL}}^\star(\delta) := \sup\left\{\pi(a^\star): \ D_{\mathrm{KL}}(\pi\|\pi_0) \leq \varepsilon\right\}.$$

**Part 1: $\chi^2$ gives $p_{\chi^2}^\star(\delta) = \Theta(\sqrt{\delta})$.**

*Upper bound.* For any $\pi$ with $\pi(a^\star) = p$,

$$\chi^2(\pi\|\pi_0) = \sum_a \frac{(\pi(a) - \pi_0(a))^2}{\pi_0(a)} \geq \frac{(\pi(a^\star) - \pi_0(a^\star))^2}{\pi_0(a^\star)}$$
$$= \frac{(p-\delta)^2}{\delta}.$$

Thus $\chi^2(\pi\|\pi_0) \leq \varepsilon$ implies $(p-\delta)^2 \leq \varepsilon\delta$, i.e.

$$p \leq \delta + \sqrt{\varepsilon\delta} = O(\sqrt{\delta}) \qquad (\delta \to 0).$$

*Lower bound.* Consider the policy that changes only the mass on $a^\star$ and keeps the remaining mass uniform:

$$\pi(a^\star) = p, \qquad \pi(a_i) = \frac{1-p}{k}, \quad i = 1, \ldots, k.$$

Let $p = \delta + c\sqrt{\delta}$ for a constant $c > 0$ (and $\delta$ small enough so that $p \leq 1$). Then the contribution of $a^\star$ to $\chi^2(\pi\|\pi_0)$ is

$$\frac{(p-\delta)^2}{\pi_0(a^\star)} = \frac{c^2\delta}{\delta} = c^2.$$

Each common action satisfies $\pi_0(a_i) = (1-\delta)/k$ and

$$\pi(a_i) - \pi_0(a_i) = \frac{1-p}{k} - \frac{1-\delta}{k} = -\frac{p-\delta}{k} = -\frac{c\sqrt{\delta}}{k},$$

so the total contribution of the $k$ common actions is

$$\sum_{i=1}^k \frac{(\pi(a_i) - \pi_0(a_i))^2}{\pi_0(a_i)} = k \cdot \frac{(c^2\delta/k^2)}{(1-\delta)/k} = \frac{c^2\delta}{1-\delta} = o(1).$$

Hence $\chi^2(\pi\|\pi_0) = c^2 + o(1)$ as $\delta \to 0$. Choosing $c < \sqrt{\varepsilon}$ makes $\chi^2(\pi\|\pi_0) \leq \varepsilon$ for all sufficiently small $\delta$, and thus

$$p_{\chi^2}^\star(\delta) \geq \delta + c\sqrt{\delta} = \Omega(\sqrt{\delta}).$$

Combining upper and lower bounds yields $p_{\chi^2}^\star(\delta) = \Theta(\sqrt{\delta})$.

**Part 2: forward KL gives $p_{\mathrm{KL}}^\star(\delta) = \Theta(1/\log(1/\delta))$.**

*Upper bound.* We consider the forward KL divergence

$$D_{\mathrm{KL}}(\pi\|\pi_0) = \sum_a \pi(a) \log \frac{\pi(a)}{\pi_0(a)}.$$

Since the objective depends only on the probability mass $p = \pi(a^\star)$ assigned to the rare action, we may group all remaining actions into a single outcome. By the grouping (data-processing) property of $f$-divergences, the problem reduces to a two-point (Bernoulli) divergence with

$$\pi = (p, 1-p), \qquad \pi_0 = (\delta, 1-\delta).$$

Thus

$$D_{\mathrm{KL}}(\pi\|\pi_0) = p \log \frac{p}{\delta} + (1-p) \log \frac{1-p}{1-\delta}.$$

If $p$ were bounded away from zero, then the first term $p\log(p/\delta)$ would diverge as $\delta \to 0$, violating the constraint $D_{\mathrm{KL}}(\pi\|\pi_0) \leq \varepsilon$. Hence any feasible sequence satisfies $p \to 0$ as $\delta \to 0$.

In this regime, the second term satisfies

$$(1-p) \log \frac{1-p}{1-\delta} = o(1),$$

while

$$p \log \frac{p}{\delta} = p \log \frac{1}{\delta} + p \log p.$$

Since $p \to 0$, we have $p \log p = -p \log(1/p) = o(p \log(1/\delta))$. Therefore,

$$D_{\mathrm{KL}}(\pi \| \pi_0) = p \log \frac{1}{\delta}\big(1 + o(1)\big).$$

It follows that if $D_{\mathrm{KL}}(\pi \| \pi_0) \le \varepsilon$, then

$$p \le \frac{\varepsilon}{\log(1/\delta)}\big(1 + o(1)\big),$$

and hence $p_{\mathrm{KL}}^\star(\delta) = O(1/\log(1/\delta))$.

*Lower bound.* Again consider the two-point distribution with $p = c/\log(1/\delta)$ for a constant $c > 0$. Substituting into the expression above yields

$$p \log \frac{p}{\delta} = c + o(1), \qquad (1-p)\log \frac{1-p}{1-\delta} = o(1),$$

and hence

$$D_{\mathrm{KL}}(\pi \| \pi_0) = c + o(1).$$

Choosing $c < \varepsilon$ ensures that the KL constraint is satisfied for all sufficiently small $\delta$, giving

$$p_{\mathrm{KL}}^\star(\delta) = \Omega(1/\log(1/\delta)).$$

Combining upper and lower bounds yields

$$p_{\mathrm{KL}}^\star(\delta) = \Theta\left(\frac{1}{\log(1/\delta)}\right).$$

$\square$

## B.8. Proof of Proposition 6.1

*Proof.* Recall

$$M^{-1} := 1/\sup_s \frac{d_{\pi^+}(s)}{d_{\pi_0}^+(s)} = \inf_s \frac{d_{\pi_0}^+(s)}{d_{\pi^+}(s)}.$$

Recall also that $\Delta = \ell(\hat{\pi}) - \ell(\pi_+)$. We write,

$$\ell(\hat{\pi}) - \ell(\pi_+)$$
$$= \mathbb{E}_{\pi_0}\left[ -\sum_{t=1}^{T-1} \log \hat{\pi}(A_t|S_t) + \sum_{t=1}^{T} \log \pi_+(A_t|S_t) \mid R(\tau) = 1 \right]$$
$$= \mathbb{E}_{\pi_0}\left[ \sum_{t=1}^{T-1} \log\left( \frac{\pi_+(A_t|S_t)}{\hat{\pi}(A_t|S_t)} \right) \mid R(\tau) = 1 \right]$$
$$= \mathbb{E}_{\pi_0}\left[ \sum_{t=1}^{T-1} \mathbb{E}_{\pi_0}\left[ \log\left( \frac{\pi_+(A_t|S_t)}{\hat{\pi}(A_t|S_t)} \right) \mid S_t, R(\tau) = 1 \right] \mid R(\tau) = 1 \right]$$
$$\overset{(a)}{=} \mathbb{E}_{\pi_0}\left[ \sum_{t=1}^{T-1} \left( \sum_a \pi_+(a|S_t) \log\left( \frac{\pi_+(a|S_t)}{\hat{\pi}(A_t|S_t)} \right) \right) \mid R(\tau) = 1 \right]$$
$$= \mathbb{E}_{\pi_0}\left[ \sum_{t=1}^{T-1} D_{\mathrm{KL}}\left( \pi_+(\cdot|S_t) \,\|\, \hat{\pi}(\cdot|S_t) \right) \mid R(\tau) = 1 \right]$$
$$\overset{(b)}{=} \sum_s d_{\pi_0}^+(s) D_{\mathrm{KL}}\left( \pi_+(\cdot|s) \,\|\, \hat{\pi}(\cdot|s) \right)$$
$$\ge M^{-1} \sum_s d_{\pi_+}(s) D_{\mathrm{KL}}\left( \pi_+(\cdot|s) \,\|\, \hat{\pi}(\cdot|s) \right)$$
$$= M^{-1} \mathbb{E}_{\pi_+}\left[ \sum_{t=1}^{T-1} D_{\mathrm{KL}}\left( \pi_+(\cdot|S_t) \,\|\, \hat{\pi}(\cdot|S_t) \right) \right]$$
$$\overset{(c)}{=} M^{-1} D_{\mathrm{KL}}\left( P_{\pi_+}(\tau) \,\|\, P_{\hat{\pi}}(\tau) \right)$$

Equality(a) is a key step, and uses that conditioned on the state, and on the trajectory ending successfully, by definition the action-distribution in that prescribed by $\pi_+$. (See the definition in Sec. 2). Equality (b) is the definiion of the occupancy measure given in Sec. 2.1. Finally, equality (c) uses the chain rule of KL divergence.

By Pinsker's inequality,

$$\|P_{\pi_+}(\tau) - P_{\hat{\pi}}(\tau)\|_{TV} \le \sqrt{\frac{1}{2} D_{KL}(P_{\pi_+} \| P_{\hat{\pi}})} = \sqrt{M\Delta/2}.$$

Since $\rho(\pi) = P_\pi(R(\tau) = 1)$ is the probability of a measurable event under the trajectory distribution,

$$|\rho(\hat{\pi}) - \rho(\pi_+)| \le \|P_{\pi_+}(\tau) - P_{\hat{\pi}}(\tau)\|_{TV} \le \sqrt{M\Delta/2}.$$

$\square$

## B.9. Proof of Proposition 7.1

*Proof.* Fix a non-terminal state $s$. Define the (faithful and proxy) advantages

$$A_{\pi_0}(s, a) := Q_{\pi_0}(s, a) - V_{\pi_0}(s),$$
$$\widetilde{A}_{\pi_0}(s, a) := \widetilde{Q}_{\pi_0}(s, a) - \widetilde{V}_{\pi_0}(s),$$

so that $\sum_a \pi_0(a|s) A_{\pi_0}(s, a) = 0$ and $\sum_a \pi_0(a|s) \widetilde{A}_{\pi_0}(s, a) = 0$.

By the analogue of Lemma B.1 applied to the faithful reward and the proxy reward respectively, the corresponding success-conditioned policies satisfy

$$\pi_+(a \mid s) = \pi_0(a \mid s)\Big(1 + \frac{A_{\pi_0}(s,a)}{V_{\pi_0}(s)}\Big),$$

and

$$\widetilde{\pi}_+(a \mid s) = \pi_0(a \mid s)\Big(1 + \frac{\widetilde{A}_{\pi_0}(s,a)}{\widetilde{V}_{\pi_0}(s)}\Big).$$

We first compute the advantage of $\widetilde{\pi}_+$ under the faithful reward:

$$
\begin{aligned}
A_{\pi_0}(s,\widetilde{\pi}_+) &:= \mathbb{E}_{a\sim\widetilde{\pi}_+(\cdot|s)}[A_{\pi_0}(s,a)]\\
&= \sum_a \widetilde{\pi}_+(a|s)\,A_{\pi_0}(s,a)\\
&= \sum_a \pi_0(a|s)\Big(1 + \frac{\widetilde{A}_{\pi_0}(s,a)}{\widetilde{V}_{\pi_0}(s)}\Big)A_{\pi_0}(s,a)\\
&= \frac{1}{\widetilde{V}_{\pi_0}(s)}\sum_a \pi_0(a|s)\,\widetilde{A}_{\pi_0}(s,a)A_{\pi_0}(s,a)\\
&= \frac{\mathrm{Cov}_{a\sim\pi_0(\cdot|s)}\Big(A_{\pi_0}(s,a),\widetilde{A}_{\pi_0}(s,a)\Big)}{\widetilde{V}_{\pi_0}(s)}.
\end{aligned}
$$

Similarly,

$$
\begin{aligned}
A_{\pi_0}(s,\pi_+) &:= \mathbb{E}_{a\sim\pi_+(\cdot|s)}[A_{\pi_0}(s,a)]\\
&= \sum_a \pi_+(a|s)\,A_{\pi_0}(s,a)\\
&= \sum_a \pi_0(a|s)\Big(1 + \frac{A_{\pi_0}(s,a)}{V_{\pi_0}(s)}\Big)A_{\pi_0}(s,a)\\
&= \frac{1}{V_{\pi_0}(s)}\sum_a \pi_0(a|s)\,A_{\pi_0}(s,a)^2\\
&= \frac{\mathrm{Var}_{a\sim\pi_0(\cdot|s)}(A_{\pi_0}(s,a))}{V_{\pi_0}(s)}.
\end{aligned}
$$

We take the ratio and insert standard normalization. For shorthand notation, let $A \sim \pi_0(\cdot|s)$.

$$
\begin{aligned}
&\frac{A_{\pi_0}(s,\widetilde{\pi}_+)}{A_{\pi_0}(s,\pi_+)}\\
&= \frac{\mathrm{Cov}(A_{\pi_0}(s,A),\widetilde{A}_{\pi_0}(s,A))/\widetilde{V}_{\pi_0}(s)}{\mathrm{Var}(A_{\pi_0}(s,A))/V_{\pi_0}(s)}\\
&= \frac{\mathrm{Cov}(A_{\pi_0}(s,A),\widetilde{A}_{\pi_0}(s,A))}{\sqrt{\mathrm{Var}(A_{\pi_0}(s,A))}\sqrt{\mathrm{Var}(\widetilde{A}_{\pi_0}(s,A))}}\\
&\qquad\times \frac{\sqrt{\mathrm{Var}(\widetilde{A}_{\pi_0}(s,A))}/\widetilde{V}_{\pi_0}(s)}{\sqrt{\mathrm{Var}(A_{\pi_0}(s,A))}/V_{\pi_0}(s)}\\
&= \mathrm{Cor}_{a\sim\pi_0(\cdot|s)}\Big(A_{\pi_0}(s,a),\widetilde{A}_{\pi_0}(s,a)\Big)\cdot\sqrt{\frac{\widetilde{\mathcal{I}}_{\pi_0}(s)}{\mathcal{I}_{\pi_0}(s)}},
\end{aligned}
$$

where $\mathcal{I}_{\pi_0}(s) = \mathrm{Var}_{a\sim\pi_0(\cdot|s)}(A_{\pi_0}(s,a))/V_{\pi_0}(s)^2$ and $\widetilde{\mathcal{I}}_{\pi_0}(s) = \mathrm{Var}_{a\sim\pi_0(\cdot|s)}(\widetilde{A}_{\pi_0}(s,a))/\widetilde{V}_{\pi_0}(s)^2$. $\qquad\square$

## C. Exact Improvement in $\rho$ under Success Conditioning.

Proposition 3.2 gives the triple identity

$$\frac{\mathbb{E}_{a\sim\pi_+(\cdot|s)}[A_{\pi_0}(s,a)]}{V_{\pi_0}(s)} = \mathcal{I}_{\pi_0}(s),$$

where $\mathcal{I}_{\pi_0}(s)$ is the action-influence (Definition 3.1). Multiplying by $V_{\pi_0}(s)$ yields the statewise identity

$$\mathbb{E}_{a\sim\pi_+(\cdot|s)}[A_{\pi_0}(s,a)] = V_{\pi_0}(s)\,\mathcal{I}_{\pi_0}(s). \tag{13}$$

We now plug (13) into the performance difference lemma. In its exact form,

$$\rho(\pi) = \rho(\pi_0) + \sum_s d_\pi(s)\,\mathbb{E}_{a\sim\pi(\cdot|s)}[A_{\pi_0}(s,a)]. \tag{14}$$

(See the discussion around the performance difference lemma in Section 4.1.)

Applying (14) to $\pi = \pi_+$ and substituting (13) gives the exact improvement formula

$$\rho(\pi_+) - \rho(\pi_0) = \sum_s w(s)\,\mathcal{I}_{\pi_0}(s). \tag{15}$$

for $w(s) = d_{\pi_+}(s)\,V_{\pi_0}(s)$. This shows *policy improvement is exactly equal to a weighted sum across states of the action-influence.*

