# OpenReview forum: "Success-Conditioning as Policy Improvement: The Optimization Problem Solved by Imitating Success"
_ICML.cc/2026/Conference — ICML 2026 regular_

### Official Review · Reviewer_Uvuo · 2026-03-08

**Soundness:** 3
**Presentation:** 4
**Significance:** 2
**Originality:** 3
**Overall Recommendation:** 4
**Confidence:** 4

**Summary:**

The paper studies RL based on success conditioning: directly imitating the actions in trajectories that were successful (compared to, e.g., taking a policy gradient step). The novelty is showing that this approach is equivalent to solving a certain trust region problem for the policy. Based on this result, the authors show several characteristics of success conditioning, for example that updates can be very conservative in certain cases. Analysis is mostly based on intuitive examples and toy problems.

**Compliance With Llm Reviewing Policy:**

Affirmed.

**Final Justification:**

The rebuttal addressed my technical questions.

I remain convinced that this paper is above the bar for acceptance, but is rather light in terms of depth/mass of contributions.

**Key Questions For Authors:**

6.1 Faithful binary rewards - doesn’t this idea introduce a lot of variance? For example, if two trajectories Y(\tau_1)=Y(\tau_2) = 0.5, it is likely that  R(\tau_1)=0 and R(\tau_2)=1. How does this reflect in the success conditioning analysis?

Related to the above - how does Figure 2 look like for a finite sample (and how does it depend on the sample size)?

**Limitations:**

yes

**Strengths And Weaknesses:**

Strengths:

- The paper is very well written. I enjoyed reading it! The toy examples are well-thought, and the explanantions are intuitive.

- The problem is relevant. Success conditioning has been popular recently, and there have been several analysis of it. This paper adds to the literature in the trust-region equivalence (original and elegant!), and the intuitive analysis of different failure cases.

- The paper advances understanding of existing algorithms (does not propose new algorithms).

Weaknesses:

- Beyond Prop 4.1, the analysis remains at a somewhat shallow level. For example: the comparison with TRPO is given in an intuitive level. However, TRPO has rate results (e.g., [1]). A rate result for success conditioning (in a tabular, online version) would allow a deeper comparison. Similarly, Lemma 4.6 gives a tolerance result, but it is not clear how this tolerance is reflected in performance.

[1] Shani, Lior, Yonathan Efroni, and Shie Mannor. "Adaptive trust region policy optimization: Global convergence and faster rates for regularized mdps." Proceedings of the AAAI conference on artificial intelligence. Vol. 34. No. 04. 2020.

- The total "mass" of the results is somewhat low compared to most ICML papers. Most theory papers usually have more (and deeper) results, and algorithmic RL papers usually include experiemnts beyond toy tasks. While this is subjective, and it can be argued if "theorem counting" is relevant for acceptance, my feeling is that this paper is on the lighter side on the amount/depth of contributions.

---

> ### Author Rebuttal · Authors · 2026-03-30
>
> ## Response to Reviewer Uvuo
>
> Thank you for the thoughtful review, and especially for engaging directly with the question of whether the paper's contributions are substantial enough. I want to address that head-on.
>
> **What the paper does.** Many of the most widely deployed training pipelines — rejection sampling with SFT, STaR, Decision Transformers — all train a model to imitate behavior conditional on success. Despite enormous engineering investment in these methods, a basic question has been open: is the thing they're all aiming for even a coherent optimization objective, or just a heuristic that happens to work? Prior work (Brandfonbrener et al., Paster et al.) showed these methods can fail badly, reinforcing the view that they are unprincipled shortcuts.
>
> This paper resolves that question exactly. The target of these pipelines is, secretly, a trust-region policy improvement step — one that provably cannot degrade performance. This places SFT on successful completions in the same intellectual family as PPO and GRPO, despite looking nothing like them. Their failure mode is precisely characterized: observable conservatism, not catastrophic drift. And a single quantity (action-influence)  governs the magnitude of improvement and is cheaply estimable as a byproduct of training a success-conditioned policy.
>
> **On the depth and mass of contributions.** The paper kept getting *shorter* as I worked on it, not longer. I think the paper's brevity may make it easy to undercount what's here. There are five conceptually independent contributions:
>
> 1. The formulation of success conditioning as a unified object across SFT, STaR, Decision Transformers, and return-conditioned RL
> 2. Identification of the optimization problem it solves (Proposition 4.1)
> 3. The definition of action-influence and the triple identity equating it to policy change and relative improvement (Proposition 4.3)
> 4. The precise separation between $\chi^2$ and KL trust-region geometries, with the tolerance result for rare actions (Lemma 4.6)
> 5. An exact decomposition of return thresholding into discrimination and alignment (Proposition 6.1), which could motivate its own line of work
>
> These build on each other but are logically independent: each answers a different question.  I would argue that the paper is quite dense with new ideas, not thin. Are you willing to reconsider the significance score of 2?
>
> **On convergence rates.** You're right that a rate result would enable a deeper comparison with TRPO, and this is a natural next step. The challenge is that, unlike TRPO, the trust-region radius here is data-dependent rather than fixed, so standard analyses don't directly apply. Weak improvement per iteration is already established (Corollary 4.5); the main obstacle to a full convergence guarantee is that action-influence could collapse to zero, stalling progress without forced exploration. I view this as one of the most promising directions opened up by the paper.
>
>
> **Variance of faithful conditioning.** Thank you — this is a great question. You are absolutely correct that a naive implementation of faithful conditioning, by randomly rounding continuous rewards to binary labels, introduces unnecessary variance.
>
> It is possible to fix this issue. One approach is to create multiple copies of each trajectory, each independently assigned a binary label — effectively averaging over the rounding noise. More elegantly, one can use *soft labels*, weighting each trajectory as a "success" with weight $Y(\tau)$ and as a "failure" with weight $1 - Y(\tau)$. This can be interpreted as the fraction of those copies that would be labeled successful, avoiding binarization variance altogether.
>
> I did not focus on this issue in Section 6.1 because the purpose of that section i to show that alternative, *non-faithful* conditioning strategies can outperform even *exact* faithful conditioning — i.e., the comparison baseline is faithful conditioning with no estimation error at all. The variance from naive binarization would only make faithful conditioning look worse, strengthening the point.
>
>
> **A finite-sample version of Figure 2:** this is a natural question, and thankfully the qualitative takeaways remain largely unchanged. I evaluate the _true policy value_ $V(\pi) = \sum_a \pi(a) Q(a)$ of policies learned from finite data, where $\pi(a) \propto \pi_0(a) \hat{\tilde{Q}}_N(a)$ is formed using empirical proxy Q-values from $N$ samples per arm. The figure ([figure](https://drive.google.com/file/d/1yB0z1vfwMSLGOpaYpl_zZtM8dCwqJucx/view?usp=sharing)) shows results for $N \in \{ 25, 250 \}$ with 10th–90th percentile bands over 200 replicates, alongside the population curve and finite-sample faithful conditioning bands. The finite-sample proxy curves closely track the population limit, and the proxy advantage over faithful conditioning is preserved at all sample sizes.

---

> > ### Author Rebuttal · Reviewer_Uvuo · 2026-04-01
> >
> > Thanks for answering my questions, and for the finite sample results.
> > My concerns about the depth/mass of contributions remain, and I do not expect them to be addressed in a short rebuttal. Overall my score accurately reflects my position on this paper.

---

> > > ### Author Response · Authors · 2026-04-02
> > >
> > > I likely have different tastes regarding the brevity of the paper vs "total mass" of the results, but I appreciate your position as well. Thanks for taking the time to carefully read and engage with both the paper and my rebuttal.

---

### Official Review · Reviewer_YenL · 2026-03-12

**Soundness:** 3
**Presentation:** 4
**Significance:** 4
**Originality:** 3
**Overall Recommendation:** 5
**Confidence:** 2

**Summary:**

This paper analyzes the optimization problem behind the Imitating success optimization strategy applied in many RL settings for improving the policy. The work proposes a theoretical analysis showing the relation with other common techniques as PPO and TRPO, based on trust regions.

**Compliance With Llm Reviewing Policy:**

Affirmed.

**Final Justification:**

I believe this work contains a strong theoretical contribution in assessing the effectiveness of Success-Conditioning, which is a widely adopted technique in RL (especially for training LLMs).

I had some minor doubts mainly related to the applicability of the proposed theoretical analysis, but I received enough clarification in the rebuttal phase.

**Key Questions For Authors:**

1. Given the proposed study, what are the domains of application best suited for applying the Imitating Success approach?
2. Do results hold also for approximation via approximate policies and finite data (as in real practical settings)?

**Limitations:**

yes

**Strengths And Weaknesses:**

# Strengths:
1. The paper is well written and all of the concepts are clearly explained. Examples help the reader in following the core ideas and understanding
2. The idea is sound and the analyses are original. This work gives a unified perspective of different techniques applied in many different settings
3. The authors provide a rigorous analysis and new theoretical insights, such as explaining why success conditioning may cause failures
4. The proposed analysis has also practical implications in RL (e.g., on LLM fine-tuning)

# Weaknesses:
1. The contribution could be strengthened by defining a mechanism to exploit the analyzed underlining optimization problem and its properties in specific settings. For example, there are no benchmarks showing cases where advantages and disadvantages are clearly exposed. It is not clear whether this study helps in defining the best domains of application for Imitating Success
2. The paper could benefit from some experimental analysis
3. It is not clear whether these results hold also for approximated policies or for finite noisy data

---

> ### Author Rebuttal · Authors · 2026-03-30
>
> Thank you for your careful review, and for recognizing the work as sound, original, and clearly explained.
>
> Your two questions are closely related, so let me address them together.
>
> Today, many of the most important training pipelines — rejection sampling with SFT for RLHF, STaR for reasoning, Decision Transformers — all do the same thing: train a model to imitate behavior conditional on success. Engineers invest enormous effort making these pipelines work. But there has been a basic unanswered question: **is the thing they're all aiming for even a sensible target from a decision-making perspective?** It could have turned out that imitating successful trajectories corresponds to no coherent optimization problem,  just a heuristic that happens to work sometimes.
>
> This paper resolves that question. The idealized policy these pipelines target is *exactly* a trust-region policy improvement step — one that provably cannot degrade performance. This places SFT on successful completions and decision-transformers in the same intellectual family as PPO and GRPO, despite looking nothing like them.
>
> **On the absence of experiments.** While the paper exactly characterizes what practical pipelines are aiming for, i.e. what it means for decision-making if they were able to imitate behavior along successful trajectories. This seems like a necessary first step, but it's fair to point out that it has not done empirical evaluation of how closely current implementations approximate what they're aiming toward. I view this as a distinct question, likely sensitive to specific algorithmic choices, and an exciting direction for follow-up. But it presupposes knowing what the target is, which is what this paper establishes. A first theoretical breakthrough in this area should, I think, be allowed to stand on its own.
>
> **On approximate policies and finite data (Q2).** The paper characterizes what these pipelines are *aiming for*. Whether they reach it in practice depends on finite-data and approximation effects, but Proposition 5.1 provides a sanity check, and notably the distribution-shift factor M equals 1 exactly in the LLM setting (deterministic dynamics), so the supervised objective aligns directly with deployment.
>
>
>
> **On best domains of application (Q1).** The theory predicts that success conditioning is well-suited to domains where action-influence is high, i.e. where the agent's choices meaningfully affect outcomes. LLM reasoning is a natural example: token choices determine whether a chain of thought reaches the correct answer. It is poorly suited when outcomes are driven mostly by environment randomness rather than action selection, as in recommender systems or advertising. Training a success-conditioned policy gives a free estimate of action-influence (see response to Reviewer PZvf for a concrete procedure), which could help practitioners diagnose this in advance.
>
> **Overall comment.** All four reviewers recognized this paper as novel, correct, and clearly written — a degree of consensus that is relatively uncommon. The limitations you mention around the lack of experiments are certainly fair, but I personally feel it's sensible for an 8-page paper to resolve what the training objectives of SFT pipelines, decision-transformers, etc. are actually *aiming for* in terms of a decision-making objective, while leaving thorough empirical investigations to future work. If you agree that the paper advances the literature in an honest way, I would appreciate if you'd consider raising your score.

---

> > ### Author Rebuttal · Reviewer_YenL · 2026-04-02
> >
> > I thank the author for the explanations, which clarify my main questions. I will update my scores accordingly

---

> > > ### Author Response · Authors · 2026-04-02
> > >
> > > Thanks a lot for taking to the time to consider the rebuttal and engage with the paper again. I really appreciate it!

---

### Official Review · Reviewer_mfP5 · 2026-03-12

**Soundness:** 4
**Presentation:** 3
**Significance:** 3
**Originality:** 4
**Overall Recommendation:** 5
**Confidence:** 3

**Summary:**

The paper studies the theoretical foundations of success conditioning, a widely used heuristic for policy improvement in RL and large-scale machine learning pipelines. In this approach, trajectories generated by a behavior policy are filtered according to whether they achieve a desired outcome, and the policy is updated by imitating the actions taken along successful trajectories. The technique appears in several modern settings, including rejection sampling with supervised fine-tuning for language model alignment, goal-conditioned reinforcement learning, and return-conditioned sequence modeling methods such as Decision Transformers. Despite its prevalence, the optimization objective implicitly solved by this procedure has not previously been characterized formally.

A core problem investigated in this work is determining what optimization principle success conditioning corresponds to, and whether the procedure can be understood as a principled policy improvement operator. The authors show that exact success conditioning yields the optimal solution to a specific trust-region policy optimization problem. In particular, the success-conditioned policy maximizes a first-order approximation to policy improvement subject to a constraint on the $\chi^2$ divergence between the new policy and the behavior policy. Unlike classical trust-region methods such as TRPO or PPO, the radius of the trust region is not chosen by the practitioner but is instead determined automatically by a quantity termed action-influence, which measures how variability in the behavior policy’s action selection affects success probability.

Overall, this study's key aspect is the establishment of an exact identity linking three quantities: relative policy improvement, the $\chi^2$ divergence between the behavior policy and the success-conditioned policy, and the action-influence under the behavior policy. The authors prove that these quantities are equal state-by-state, implying that success conditioning behaves as a conservative policy improvement operator. The paper further analyzes the failure modes of success conditioning, demonstrating that the method fails primarily through excessive conservatism (making minimal policy updates when action-influence is small) rather than through instability or catastrophic policy degradation. Additional results examine the effects of proxy rewards such as return thresholding and provide a bound relating deployment performance to the supervised learning loss incurred when approximating the success-conditioned policy from finite data.

**Compliance With Llm Reviewing Policy:**

Affirmed.

**Final Justification:**

Authors addressed concerns.

**Key Questions For Authors:**

1) The theory highlights action-influence as the central quantity governing improvement. In realistic settings where $Q_{\pi_0}(s,a)$ is unknown, how do the authors recommend estimating or approximating this quantity in practice? Would approximate value estimation undermine the theoretical interpretation?

2) Proposition 5.1 bounds deployment performance using excess cross-entropy loss and a distribution-shift factor $M$. In realistic neural policy classes, how large might $M$ be expected to become? Are there practical regimes where the bound is likely to be informative?

3) The paper briefly mentions iterated success conditioning as a future direction. Do the authors have conjectures regarding the convergence properties of repeated application of the operator $\pi^+$? For instance, does repeated success conditioning converge to a fixed point or optimal policy under reasonable assumptions?

4) Many offline RL algorithms update policies via exponential weighting of actions according to estimated advantages. Can the authors more precisely characterize when success conditioning approximates such exponential updates, or when the two approaches diverge substantially?

5) The paper suggests that the $\chi^2$ divergence between $\pi_0$ and $\pi^+$ may serve as a diagnostic for improvement. Have the authors observed this relationship empirically in practical training pipelines, such as rejection sampling with supervised fine-tuning in language models?

**Limitations:**

Yes.

**Strengths And Weaknesses:**

**Strengths:**
The paper addresses an important and timely question: how to formally understand the objective implicitly optimized by widely used success-conditioning techniques in RL and language model training. The central contribution is a proof that success conditioning exactly solves a trust-region policy optimization problem with $\chi^2$ divergence and a data-determined radius. This is both technically interesting and conceptually clarifying. The results establish a surprising equivalence between Bayesian conditioning on success events and constrained policy improvement, providing a principled lens through which existing methods such as rejection sampling with supervised fine-tuning, return-conditioned sequence modeling, and related approaches can be interpreted. The theoretical development is elegant, particularly the identity equating relative advantage, $\chi^2$ divergence, and action-influence, which yields a clear interpretation of the magnitude of policy updates and their relationship to achievable improvement. The analysis of proxy rewards and return thresholding is also insightful, offering a precise characterization of the tradeoff between increased discrimination and potential objective misalignment. From a theoretical perspective, the work is largely self-contained, relying on classical results such as the performance difference lemma while introducing novel quantities and identities that deepen understanding of policy improvement mechanisms.

**Weaknesses:**
Despite the appealing theoretical characterization, several aspects of the work limit its practical and conceptual impact. First, the main results concern exact success conditioning under a fixed behavior policy and assume access to the exact success-conditioned distribution. In realistic large-scale settings, such as language model training or offline RL, policies are only approximate solutions to the supervised objective and are learned from finite datasets with significant function approximation error. While Proposition 5.1 provides a deployment bound relating excess cross-entropy loss to performance degradation, this result relies on a distribution-shift factor that may be large in general environments. As a result, the paper leaves open whether the theoretical guarantees meaningfully translate to the regimes where success conditioning is currently used in practice. Second, the theoretical framework assumes a binary success indicator and episodic MDP formulation. Although the authors discuss a reduction from dense rewards to binary rewards via stochastic labeling, it is unclear whether this reduction meaningfully captures the optimization objectives used in many RL or language model training pipelines. The implications of the results for these broader settings remain somewhat indirect. Third, the paper does not include empirical validation. While the authors emphasize that the results are theoretical equalities rather than empirical claims, a small set of experiments illustrating the practical relevance of action-influence, the predicted conservatism of updates, or the diagnostic role of the $\chi^2$ divergence would substantially strengthen the paper. Without empirical evidence, it is difficult to assess whether the proposed interpretation provides actionable guidance for practitioners. Lastly, some assumptions underlying the analysis deserve more explicit discussion. For example, the theory implicitly assumes that success occurs with positive probability at all states and that the behavior policy sufficiently explores the relevant action space. In environments where these assumptions fail, particularly in offline RL with limited support, the behavior of success conditioning could differ substantially from the theoretical predictions. Clarifying these limitations would help readers better understand the scope of the results.

---

> ### Author Rebuttal · Authors · 2026-03-30
>
> ## Response to Reviewer mfP5
>
> Thank you for a very thorough review. I truly appreciated the comments — that the paper "addresses an important and timely question," "establishes a surprising equivalence," with analysis that is "elegant" and "insightful," "introducing novel quantities and identities that deepen understanding of policy improvement mechanisms." You rated Soundness 4 (excellent) and Originality 4 (excellent). Let me address your questions concisely, noting that some points overlap with responses to other reviewers. I hope this addreses your remaining concerns.
>
> **On the binary reward assumption.** Conditioning on success inherently requires a binary criterion. That said, Section 6 addresses the most common way success conditioning is applied with dense rewards — return thresholding — and provides an exact decomposition of its effects into discrimination (action-influence amplification) and alignment with the true objective (Proposition 6.1). Could you clarify what practical objectives you feel this does not capture?
>
> **Q1: Estimating action-influence in practice.** A key point is that training a success-conditioned policy yields, almost for free, an estimate of action-influence. By Proposition 4.3, action-influence equals the χ² divergence between π₊ and π₀, which can be estimated from a hold-out set of successful trajectories using only forward passes through both models. See the response to Reviewer PZvf for a concrete estimator with pseudocode. Crucially, this avoids approximate value estimation.
>
> **Q2: The distribution-shift factor M and practical relevance.** Many training pipelines — SFT, STaR, Decision Transformers — all approximate a common idealized object: a policy imitating behavior conditional on success. The paper asks whether that object is sensible to target, and resolves the question exactly. Proposition 5.1 is then a sanity check, showing that as pipelines become better at minimizing the imitation learning loss they are trained to minimize, the paper's theory offers increasingly accurate predictions of the improvement offered by success conditioning. The distribution-shift factor M is common in RL theory and often environment-dependent in a worrying way, but here $M = 1$ exactly under deterministic dynamics — which includes autoregressive language generation — so there is no distribution-shift penalty in the setting where success conditioning is most widely used.
>
> **Q3: Iterated success conditioning.** The theory already shows that repeated application yields a sequence of policies that weakly improve at each iteration (Corollary 4.5). In every non-degenerate numerical example I have constructed, it also strictly converges to optimality. The main obstacle to a general convergence guarantee is the lack of forced exploration — the policy could stabilize at a suboptimal fixed point if action-influence collapses to zero. I know of one researcher who has made substantial progress on this question, inspired by this paper.
>
> **Q4: Comparison with exponential weighting.** By Lemma A.1, success conditioning reweights actions proportionally to $Q(s,a)/V(s)$, while exponential methods use weights proportional to $\exp(\beta A(s,a))$. When advantages are small relative to values, these are approximately first-order equivalent, but they diverge for large advantages — exponential weighting is more aggressive in upweighting high-advantage actions, while the χ² geometry of success conditioning is more conservative. Sections 4.5.1 and 4.5.2 discuss the differences in detail, though the primary contribution is showing that success conditioning is, surprisingly, in the same *family* as PPO and GRPO.
>
> **Q5: Empirical observation of the χ² diagnostic.** See the response to Reviewer PZvf for a concrete procedure to estimate this quantity. The mathematical theory says that insofar as a training pipeline successfully minimizes the imitation learning loss it aims to minimize, the magnitude of policy improvement is estimated in a simple, unbiased way by that procedure. The theory would not apply if an implementation badly fails to approximate the true success-conditioned policy. I am eager to do empirical evaluations with existing implementations, but view that as orthogonal to this submission.
>
> **On the scope of assumptions.** You note that the theory assumes success occurs with positive probability at all states. This is stated explicitly in Section 2 and is necessary for the success-conditioned policy to be well-defined — it is a regularity condition rather than a substantive restriction. In offline RL with limited support, success conditioning would indeed behave differently, but this is a setting where the method is not typically applied; the dominant use cases (LLM fine-tuning, reasoning) involve on-policy data where the assumption holds naturally.

---

> > ### Author Rebuttal · Reviewer_mfP5 · 2026-04-02
> >
> > Thanks!

---

> > > ### Author Response · Authors · 2026-04-02
> > >
> > > Thanks a lot for taking to the time to consider the rebuttal and engage with the paper again. I really appreciate it!

---

### Official Review · Reviewer_9Zvf · 2026-03-13

**Soundness:** 3
**Presentation:** 4
**Significance:** 3
**Originality:** 3
**Overall Recommendation:** 5
**Confidence:** 4

**Summary:**

This work studies the success conditioning policy improvement technique in a theoretical setting. Success conditioning, primarily targeting sparse reward MDPs (with reward 0 / 1 given only in terminal states), is the practice of fitting a policy to imitate the behavior policy $\pi_0$ when conditioned on success, i.e., maximize the log-likelihood of actions conditioned on states taken along successful trajectories.
The main result characterizes success conditioning as solving a trust region policy optimization problem, where the linearization of the value is maximized subject to the policy remaining within a region defined by the success conditioned occupancy measure of $\pi_0$, chi-squared divergence from $\pi_0$ and its **action influence**; roughly, at a given state - the expectation (over $a\sim \pi_0$) of the (advantage normalized by the value)-squared.

The weighted action influence identity relates the relative 1st-order improvement to the weighted action influence and magnitude of the change; and is a direct result of the per state identity: relative advantage=chi-squared divergence=action influence.

The paper continues to discuss relations with well known trust region based policy optimization algorithms, and presents extensions to the dense reward setting.

**Compliance With Llm Reviewing Policy:**

Affirmed.

**Final Justification:**

I didn't have any major concerns that warranted a response from the authors.
My one concern was with regards to the depth of the technical contribution, but as I originally stated I don't view this as a major weakness. Hence my position remains the same, favoring clear acceptance.

**Key Questions For Authors:**

- Line 226 "Among these three equal quantities, the magnitude of policy change is particularly useful as a diagnostic." / "By contrast, the χ 2 divergence between π+ and π0 is directly observable during training." - There is not discussion regarding how to estimate the quantities in question. How well is chi-squared-divergence estimable from data? Would a useful, reliable procedure be feasible in an RLVR setting where this is commonly adopted?  Are there other practical settings where this may be applicable? In was also wondering if maybe the deterministic MDP setting makes things easier in some sense, given that $d^+_\pi = d_{\pi^+}$.
  Where limitations section provides a disclaimer for the claim in bold in question; I would suggest bringing some of that discussion in the same place where you make this claim, and / or remove the bold font.
- Since language modeling and reasoning are discussed as motivators, I would discuss the fact that for deterministic dynamics  $d^+_\pi = d_{\pi^+}$ earlier. Noting that in general this is not true but for deterministic dynamics it is would elucidate these notions for a first time reader, which would be wondering about this on their own already when the notation is introduced.
- Typos
	- "The right measure of this turns out [to] be the measure below"
	- "The success-conditioned policy π+ is an optimal solution to the trust region problem (1) with" - what is (1)? broken reference
	- "Alternatives to TRPO based on forward KL or entropy regularized objectives have also been studied"
	- I think your usage of 'forward' and 'reverse' kl is flipped, forward is when the true (=target, non-fitted) distribution is the first argument.
	- Figure 1 needs adjustments (title of the bottom plot overlaps with x axis label of top plot)

**Limitations:**

Yes

**Strengths And Weaknesses:**

### Strengths
- This work studies a popular technique and establishes a nice characterization, that clarifies (to a degree) what success conditioning is actually doing. It definitely deepens our understanding of this algorithm.
- The paper is well written and easy to follow.
- There may be useful implications for practical setups. In particular, Proposition 4.3 suggests that  estimating the magnitude of policy change provides a good estimate of the relative first order improvement obtained by success conditioning, which in itself is not easily estimable from the offline data. (However, I think that perhaps a more elaborate discussion is warranted - see my comments below.)

### Weaknesses

- There is not much technical depth nor are there experiments that can demonstrate any practical implications. At the same time I will say that I view the theoretical contributions to stand on their own right, and it is not necessarily "bad" that these were obtained without much technical challenge.

---

> ### Author Rebuttal · Authors · 2026-03-30
>
> Thank you for a careful reading of the paper and thoughtful comments. Thank you also for catching several typos and more minor issues, which I will address.
>
> > There is not much technical depth nor are there experiments that can demonstrate any practical implications. At the same time I will say that I view the theoretical contributions to stand on their own right, and it is not necessarily "bad" that these were obtained without much technical challenge.
>
> This is certainly a reasonable comment. It may be worth mentioning that a ton of work went into formulating the problem, asking the questions, stating results, and structuring proofs in a very clean way such that, in hindsight, nothing seems burdensome. Upfront, it was not so clear how to think about various occupancy measures, divergence measures, the way return thresholding optimizes a proxy reward etc. At a higher level, it also wasn't clear at the outset whether the widely-used methods I call 'success conditioning' correspond to any coherent optimization problem being solved, or whether this is simply a heuristic with no crisp interpretation within RL frameworks.
>
>
>
> > - Since language modeling and reasoning are discussed as motivators, I would discuss the fact that for deterministic dynamics $d^{+}_{\pi} = d_{\pi^+}$ earlier.
>
> This is a great suggestion! Truthfully the difference between this is quite subtle. The conciseness of the proofs hides how easy it is to make mistakes with these.  Concretely, I'll note that the $d^{+}_{\pi} = d_{\pi^+}$ in the LLM setting when the notation is introduced, and that this is precisely what makes M=1 in Proposition 5.1 for the that setting.
>
> **Estimating aggregate action-influence from data**.
>
> > There is not discussion regarding how to estimate the quantities in question. How well is chi-squared-divergence estimable from data? Would a useful, reliable procedure be feasible in an RLVR setting where this is commonly adopted? Are there other practical settings where this may be applicable?
>
> There is actually a very clean procedure, I agree it would be helpful to describe this in more detail. Here are the details
>
> By Proposition 4.3, the aggregate action-influence equals the weighted $\chi^2$ divergence between $\pi_+$ and $\pi_0$. Expanding the definition of $\chi^2$ divergence, this can be rewritten as
>
> $$\mathbb{E}\left[\sum_{t=1}^{T-1} (\frac{\pi_+(A_t \mid S_t)}{\pi_0(A_t \mid S_t)} - 1)^2 \right] $$
>
> where $(S_1, A_1, \ldots, S_T) \sim P_{\pi_0}(\tau \mid R(\tau) = 1)$ is a successful trajectory from the base policy $\pi_0$. The key observation is that this is the expectation of a single scalar — the trajectory-level sum of squared likelihood-ratio deviations — so it concentrates at the usual rate given a modest hold-out set of successful trajectories from $\pi_0$.
>
> Given access to both the base model and a success-conditioned one (e.g. post-trained by SFT), the estimator requires only evaluating each model's log-probability at the token actually taken along each held-out trajectory:
>
> ```python
> # Inputs: base model π₀, success-conditioned model π₊
> # Hold-out: N successful trajectories from π₀
> estimates = []
> for τ in successful_trajectories:
>     traj_sum = 0
>     for (s_t, a_t) in state_action_pairs(τ):
>         r = π₊(a_t | s_t) / π₀(a_t | s_t)
>         traj_sum += (r - 1)**2
>     estimates.append(traj_sum)
>
> mean = avg(estimates)
> se = std(estimates) / sqrt(N)    # → immediate confidence intervals
> ```
>
> This involves just one forward pass through each model per trajectory. The procedure applies identically to any RL problem with discrete actions where both policies can be queried at observed state-action pairs.

---

> > ### Author Rebuttal · Reviewer_9Zvf · 2026-04-03
> >
> > Thank you for the detailed reply. I think this is nice work and remain in favor of acceptance.
> >
> > **1.**
> > I remember I had an explanation for this but am now not sure again. Regarding Line 226 "Policy
> > improvement is the object of interest but is only revealed after deployment." Why can we not measure this quantity when we are done training? It only involves the linearized objective (which we are optimizing) and the value of $\pi_0$, which we can estimate. With this in mind, why should we prefer estimating the chi-squared divergence?
> >
> > **2.**
> > One thought that came to mind while reading the paper, which might be worth a discussion, was that success conditioning (SC) in the RLVR setup (0/1 reward, horizon 1, deterministic dynamics) is pretty similar to the recently proposed maximum-likelihood objective (MaxRL [1]; see also [2] which is similar in spirit).
> > It seems that at the prompt level, these are both the same, but they differ when considering the full dataset. SC takes expectation over prompts from the success conditioned posterior, while MaxRL from the prior.
> > Nonetheless, if I am not mistaken your arguments would work equally well over the prior (in the RLVR setting), in which case you obtain a different formulation for the MaxRL objective, interestingly coming back to an RL trust region problem.
> >
> > Further, Lagrangian duality should allow reframing this as a regularized objective where the $\pi_0$ action influence term can be dropped as it is constant (this should be also true for SC).
> > Now, as it turns out, MaxRL can be characterized by a standard regularized policy optimization step, but with chi-square regularization in place of the standard KL one. Or equivalently, as the optimal policy in a chi-squared regularized MDP.
> >
> >
> > *Disclaimer: I didn't check this carefully.
> >
> >
> >
> > [1] Tajwar, F., Zeng, G., Zhou, Y., Song, Y., Arora, D., Jiang, Y., Schneider, J., Salakhutdinov, R., Feng, H., & Zanette, A. Maximum Likelihood Reinforcement Learning. arXiv preprint arXiv:2602.02710, 2026.
> >
> > [2] Osband, I., 2026. Delightful Distributed Policy Gradient. arXiv preprint arXiv:2603.20521.

---

> > > ### Author Response · Authors · 2026-04-03
> > >
> > > Thank you for the continued engagement. These comments are extremely helpful - wow!
> > >
> > > **Q1: Why prefer the $\chi^2$ diagnostic over direct measurement of policy improvement?**
> > >
> > > Unlike explicit policy optimization procedures, success conditioning never estimates a value function or constructs a decision-making objective — those objects appear only implicitly in the theory. The pipeline takes in a behavior policy, filters for successful trajectories, and outputs a new policy. Moreover, these methods are typically studied in an offline RL setting, where there is no simulator to directly roll out and evaluate the new policy. So "measure policy improvement when we are done training" is less straightforward than it sounds —$L_{\pi_0}(\pi_+)$ involves advantages and values that success conditioning deliberately avoids computing.
> > >
> > > The $\chi^2$ divergence sidesteps this entirely. It measures the magnitude of change from $\pi_0$ to $\pi_+$, which aligns with the objects that a success conditioning pipeline (or a Decision Transformer) actually estimates — just the two policies. The action-influence identity then tells you that this directly observable quantity *is* the relative first-order improvement, without ever needing to estimate values. I'll clarify this distinction in the revision.
> > >
> > > **Q2: Connection to MaxRL.**
> > >
> > > Thanks for raising this. It looks like this paper appeared on arxiv after I submitted to ICML, and I hadn't read it, but now I think the connection is quite illuminating. You're right that at the per-prompt level, MaxRL and success conditioning perform essentially the same update: imitating successful trajectories. The key difference is in aggregation across prompts. If I understand correctly, MaxRL's log-likelihood objective induces a weighting that upweights hard prompts — so the weighting over prompts isn't quite the prior, but something closer to the inverse success probability, related to the probability of having at least one success across N generations. That intuition seems also to relate to Osband's paper.
> > >
> > > Your observation here is intriguing. If the trust-region formulation from my paper can be recast as a $\chi^2$-regularized objective (which, morally, should hold), then MaxRL's per-prompt update would correspond to an optimal policy step in a certain $\chi^2$-regularized MDP.  This is quite a cool connection! I'll work through this argument to get all of the details right.

---

### Decision · Program_Chairs · 2026-04-30

**Decision:**

Accept (regular)

**Comment:**

Success conditioning—learning to act as you would have, given that you eventually succeeded—has emerged as a popular and general technique in policy learning and a simple alternative to reinforcement learning or a heuristic improvement over standard behavior cloning, which is otherwise agnostic to how good decisions may be. This submission provides an analysis of the success conditioning strategy, which, despite its popularity, is still poorly understood.

Reviewers appreciated the elegance of the theoretical analysis, the clear writing, the importance of the topic, and the timeliness of the submission. Concerns were raised about the technical depth, and the reviewers asked questions about the plausibility of the assumptions placed on the analyzed policies in real-world scenarios. Generally, the rebuttal was well-received and, following it, all reviewers recommended the acceptance of the paper. All in all, the submission would make a very nice contribution to the program.